# Mechanisms of opening and closing of the bacterial replicative helicase

Jillian Chase[1,2], Andrew Catalano[1], Alex J Noble[3], Edward T Eng[3], Paul DB Olinares[4], Kelly Molloy[4], Danaya Pakotiprapha[5,6], Martin Samuels[6], Brian Chait[4], Amedee des Georges[1,2,7,8]*, David Jeruzalmi[1,2,8,9]*

[1]Department of Chemistry and Biochemistry, City College of New York, New York, United States; [2]PhD Program in Biochemistry, The Graduate Center of the City University of New York, New York, United States; [3]Simons Electron Microscopy Center, The New York Structural Biology Center, New York, United States; [4]Laboratory for Mass Spectrometry and Gaseous Ion Chemistry, The Rockefeller University, New York, United States; [5]Department of Biochemistry, Center for Excellence in Protein and Enzyme Technology, Faculty of Science, Mahidol University, Bangkok, Thailand; [6]Department of Molecular and Cellular Biology, Harvard University, Cambridge, United States; [7]Structural Biology Initiative, CUNY Advanced Science Research Center, New York, United States; [8]PhD Program in Chemistry, The Graduate Center of the City University of New York, New York, United States; [9]PhD Program in Biology, The Graduate Center of the City University of New York, New York, United States

*For correspondence:
amedee.desgeorges@asrc.cuny.
edu (AG);
dj@ccny.cuny.edu (DJ)

**Competing interests:** The authors declare that no competing interests exist.

**Abstract** Assembly of bacterial ring-shaped hexameric replicative helicases on single-stranded (ss) DNA requires specialized loading factors. However, mechanisms implemented by these factors during opening and closing of the helicase, which enable and restrict access to an internal chamber, are not known. Here, we investigate these mechanisms in the *Escherichia coli* DnaB helicase•bacteriophage λ helicase loader (λP) complex. We show that five copies of λP bind at DnaB subunit interfaces and reconfigure the helicase into an open spiral conformation that is intermediate to previously observed closed ring and closed spiral forms; reconfiguration also produces openings large enough to admit ssDNA into the inner chamber. The helicase is also observed in a restrained inactive configuration that poises it to close on activating signal, and transition to the translocation state. Our findings provide insights into helicase opening, delivery to the origin and ssDNA entry, and closing in preparation for translocation.
DOI: https://doi.org/10.7554/eLife.41140.001

## Introduction

Chromosomal replicative helicases are hexameric protein ensembles that travel ahead of the advancing replisome, dissolving duplex DNA into templates for DNA synthesis. Melting of the duplex arises from ATP dependent translocation of the helicase along single stranded (ss) DNA, paired with inclusion/exclusion of DNA strands from an internal chamber. In bacteria, replicative helicases are closed protein rings, and the mechanisms associated with their loading onto chromosomal DNA, which is effectively an infinitely long polymer with no free termini, remain to be clarified. To assemble such entities on DNA, bacteria use specialized loading factors that mediate opening of the protein ring, guiding of ssDNA into the exposed chamber, and, finally, sealing of the helicase with ssDNA trapped inside (*Soultanas, 2012*; *Bell and Kaguni, 2013*; *O'Donnell et al., 2013*; *O'Shea and Berger, 2014*; *Chodavarapu and Kaguni, 2016*; *Hauk and Berger, 2016*; *Bleichert et al., 2017*).

The architecture and mechanism of translocation of ring-shaped replicative helicases have been extensively studied (*Yu et al., 1996*; *Sawaya et al., 1999*; *Singleton et al., 2000*; *Yang et al., 2002*; *Enemark and Joshua-Tor, 2006*; *Núñez-Ramírez et al., 2006*; *Bailey et al., 2007a*; *Bailey et al., 2007b*; *Wang et al., 2008*; *Kashav et al., 2009*; *Lo et al., 2009*; *Thomsen and Berger, 2009*; *Itsathitphaisarn et al., 2012*; *Kaplan, 2013*; *Robinson and van Oijen, 2013*; *Strycharska et al., 2013*; *Lee et al., 2014*; *Bazin et al., 2015*; *Fernández-Millán et al., 2015*; *O'Donnell and Li, 2018*). Less is known, however, of opening, assembly on ssDNA, and closing of the helicase as catalyzed by helicase loaders (*Arias-Palomo et al., 2013*; *Liu et al., 2013*).

Assembly of the bacterial replicative helicase on chromosomal DNA takes place during the initiation phase of DNA replication (*LeBowitz et al., 1985*; *Learn et al., 1997*; *Stephens and McMacken, 1997*; *Fok, 2002*; *Riazuddin, 2003*; *Stepankiw et al., 2009*; *Ozaki et al., 2012*; *Bell and Kaguni, 2013*; *Chodavarapu and Kaguni, 2016*) (*Figure 1A*). Several replication initiation systems have been studied, including those that operate on the primary (*Mott and Berger, 2007*; *Wolański et al., 2014*; *Leonard and Grimwade, 2015*) and secondary chromosomes (*Egan and Waldor, 2003*; *Val et al., 2014*; *Gerding et al., 2015*; *Orlova et al., 2017*; *Fournes et al., 2018*) of bacteria, plasmids (*Konieczny et al., 2014*), and phage λ (*Weigel and Seitz, 2006*). In these systems, four molecular elements cooperate to begin the process of assembling the replisome. These elements are: a) a DNA sequence of length in the hundreds of basepairs called a replication origin, b) the replication initiator protein (*E. coli*: DnaA (*Mott and Berger, 2007*; *Wolański et al., 2014*; *Leonard and Grimwade, 2015*) *V. cholera*: DnaA, RctB (*Egan and Waldor, 2003*; *Val et al., 2014*; *Gerding et al., 2015*; *Fournes et al., 2018*), plasmids: RepE, Pi, TrfA (*Konieczny et al., 2014*), phage λ: O (*Weigel and Seitz, 2006*)), c) the replicative helicase, and d) the helicase loader. Multiple copies of the initiator protein bind to distinct sites on origin DNA and associate into a large protein nucleic acid complex that is believed to have DNA around protein. One important output of this complex is melting of an A-T rich segment termed the DNA unwinding element (DUE). The initiator protein DnaA has been shown to bind to this melted segment (*Speck and Messer, 2001*; *Duderstadt et al., 2011*; *Chodavarapu and Kaguni, 2016*; *Bleichert et al., 2017*). Unwound DNA at the origin provides an entry point for assembly of the replicative helicase, which arrives at the origin bound to the helicase loader.

Recruitment of the replicative helicase to initiator-produced ssDNA at the origin proceeds through an assembly pathway with at least four stages (*Figure 1B*). Stage I comprises the isolated hexameric DnaB helicase, which is found in two closed planar ring conformations, termed dilated and constricted (*Bailey et al., 2007a*; *Strycharska et al., 2013*); these differ on the relative orientation of subunits and the diameter of the central chamber. In Stage II, the loader captures the helicase, leading to inhibition of its ATPase and ssDNA translocation activities (*Wahle et al., 1989a*; *Wahle et al., 1989b*; *Mallory et al., 1990*; *Davey et al., 2002*). *E. coli* DnaC serves as the helicase loader and delivers the DnaB helicase to the bacterial origin; it is unrelated in sequence to λP, the

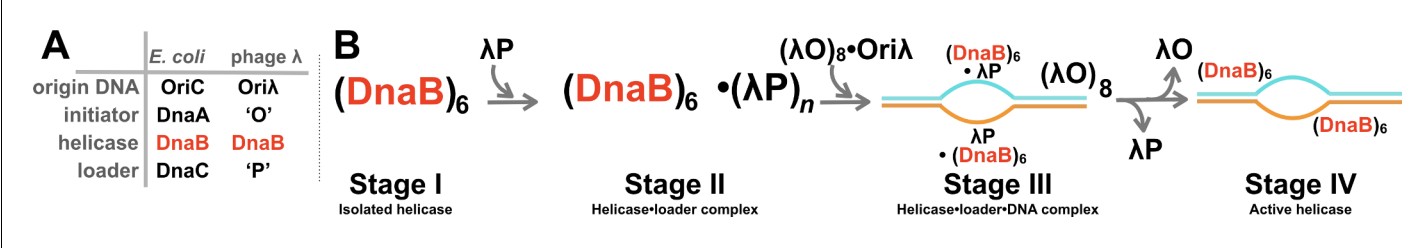

**Figure 1.** Initiation of DNA replication in bacteria and the assembly pathway for the replicative helicase. (**A**) The four core molecular entities required for the initiation of DNA replication in *E.coli* and bacteriophage λ. The phage encoded 'O' and 'P' proteins recruit the host replication apparatus to drive replication from the phage Oriλ origin. The DnaB helicase participates in the initiation of DNA replication of both the chromosomal and phage genomes. (**B**) Prior work has defined at least four stages in the assembly pathway of the hexameric DnaB bacterial replicative helicase. Stage I features the isolated helicase. In Stage II, the helicase is captured by the helicase loader. In Stage III, the helicase • loader complex engages ssDNA at the origin, which is produced by the action of the initiator protein. In Stage IV, the loader has been expelled, and the helicase assumes an active conformation, which is competent to translocate along ssDNA.
DOI: https://doi.org/10.7554/eLife.41140.002

loader that operates at the λ phage origin (*Mallory et al., 1990*), and which is the focus of this work. In Stage III, the helicase•loader complex engages ssDNA and the initiator protein at the origin. Transition to Stage IV involves expulsion of the loader from the origin complex, with concomitant activation of the helicase's enzymatic activities, and assumption of the closed spiral conformer that can translocate along ssDNA (*Itsathitphaisarn et al., 2012*).

A series of insightful analyses have shed light on the dynamic architecture of Stage I of the bacterial replicative helicase assembly pathway (*Yu et al., 1996*; *Yang et al., 2002*; *Toth et al., 2003*; *Núñez-Ramírez et al., 2006*; *Bailey et al., 2007a*; *Wang et al., 2008*; *Stelter et al., 2012*; *Strycharska et al., 2013*; *Bazin et al., 2015*) and the mechanisms of DNA unwinding by the DnaB helicase in Stage IV (*Itsathitphaisarn et al., 2012*). However, relatively little is known about Stages II and III, as well as the transitions that link each stage, where the DnaB-helicase is opened and closed with ssDNA sequestered in its internal chamber. Although low-resolution structural and biochemical analyses provided insights into the DnaB•DnaC complex (BC) (Appendix), the Stage II complex from bacteria (*Arias-Palomo et al., 2013*), important questions remain to be addressed. For example, (1) how does the helicase loader open the closed ring DnaB-helicase?, (2) how is helicase activity suppressed by the loader to prevent unwinding of DNA prior to firing of the origin?, and, (3) how does the helicase close once ssDNA has been admitted into the inner chamber, with concomitant relief of inhibition of helicase activity?

Below, we address molecular mechanisms that accompany capture of the helicase by the loader in Stage II, and the transitions that link this stage to prior and ensuing events of the loading pathway. We report on the structure of the *Escherichia coli* DnaB-helicase•bacteriophage λP helicase loader complex (henceforth: BP) by single particle cryoEM at 4.1 Å resolution. In the identified complex, we observe five λP loader molecules bound to the helicase at five consecutive DnaB subunit interfaces; the sixth DnaB interface has been breached, thus precluding a sixth λP from binding. We confirm the unanticipated $B_6P_5$ stoichiometry by native mass spectrometry. The λP loader restructures layers in the DnaB helicase comprised of carboxy-terminal (CTD) and amino-terminal (NTD) domains into novel right-handed open spiral configurations. Restructuring breaks one of the six helicase subunit interfaces to produce ~15 Å and ~20 Å openings in the CTD and NTD layers, respectively; these openings are of sufficient size to enable access by ssDNA to the internal chamber of the DnaB-helicase. Furthermore, reconfiguration forces the CTD layer of the helicase into a restrained inactive conformation wherein the ATP hydrolytic and DNA-binding properties are diminished, if not abolished. The restrained configuration of the CTD tier is poised to relax into the active conformation on expulsion of the loader from the complex. The NTD layer is also an important locus of conformational changes, which both contributes to closing of the helicase and prepares the helicase to interact with components of the replisome. Our findings reveal insights into mechanisms of opening and closing of the helicase, and provide a coherent structural view of helicase loading at the origin of DNA replication.

## Results

### Architecture and stoichiometry of the DnaB helicase • λP helicase loader complex

The structure of the BP complex was determined using cryo electron microscopy (EM) and tomography to a resolution of 4.1 Å (*Figure 2*; *Figure 2—figure supplements 1*, *2*, *3*, *4*, *5* and *6*, *Table 1*, and Appendix). Six DnaB protomers were unambiguously positioned in the EM map that we obtained. Although the EM sample contained ATP, the density maps showed that only five of six nucleotide-binding sites on CTD domains were populated, and with ADP (below and *Figure 2—figure supplement 6B*). Five copies of the λP helicase loader were also visible in the density maps. Cross-linking mass spectrometry (CX-MS), along with binding studies (*Figure 3A and B*, *Figure 3—figure supplement 1*, *Table 2* and Appendix), was used to unambiguously assign the N to C chain polarity for λP and develop a tentative assignment of the amino acid sequence to the structure.

The unanticipated $B_6P_5$ stoichiometry predicted by the density map was verified using native mass spectrometry (MS) (*Figure 3C*, *Table 3* and Appendix). Our measurements revealed a predominant species with a mass of 446.3 kDa corresponding to a $B_6P_5$ entity; additional species with masses of 472.8 kDa ($B_6P_6$) and 419.7 kDa ($B_6P_4$) were observed at lower intensities. However,

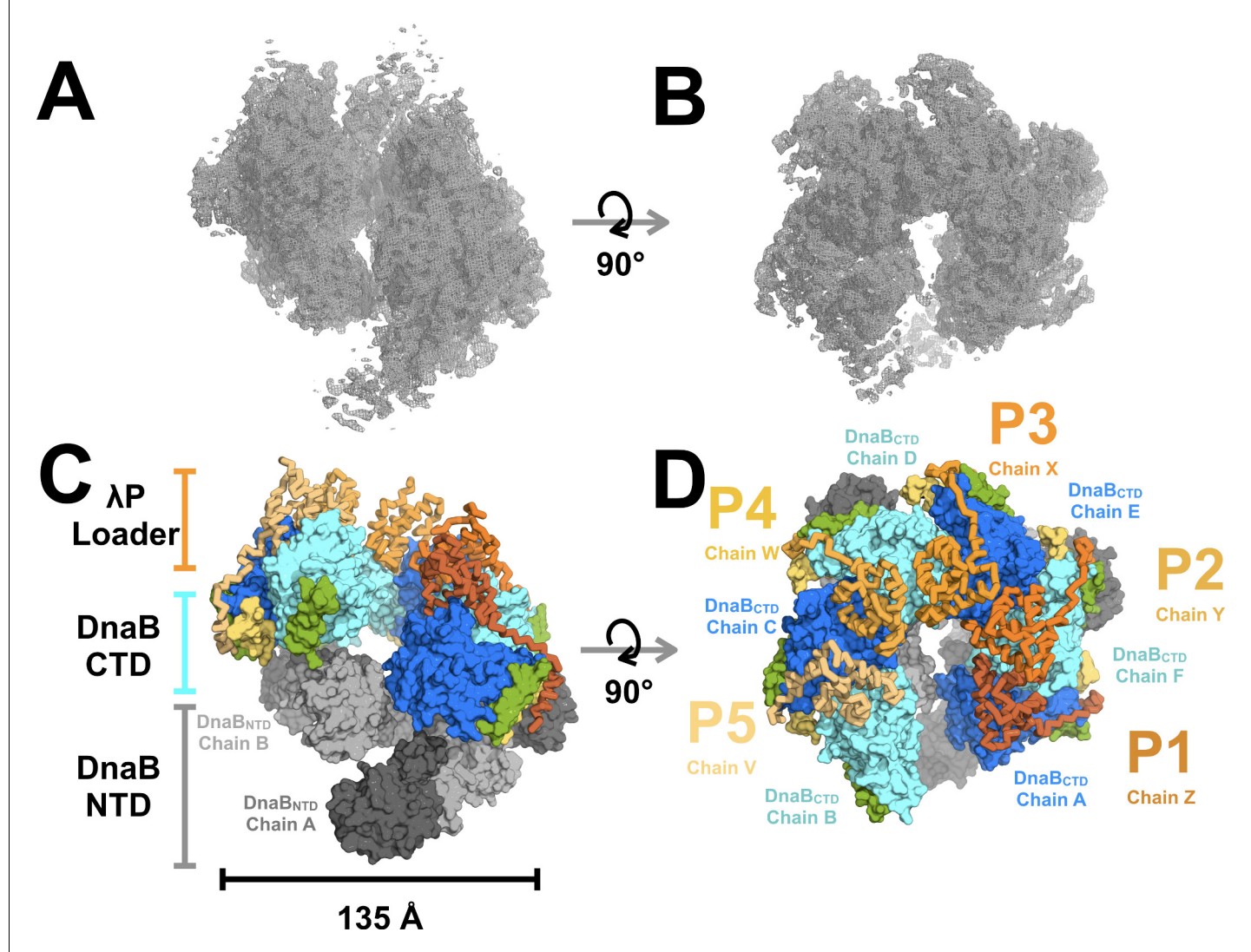

**Figure 2.** Architecture and stoichiometry of the *E.coli* DnaB helicase•bacteriophage λP complex. (**A**) Experimental EM density map of the *E. coli* DnaB helicase•bacteriophage λP complex contoured at five sigma in PyMOL (see also Supplemental *Figure 2—figure supplement 3*). (**B**) Same as panel A except that the map has been rotated by 90°. (**C**) The *E. coli* DnaB helicase•bacteriophage λP complex is shown depicting the ruptured interface between DnaB subunits A and B and the deep canyon that runs through the complex. The complex has been sub-divided into three tiers: λP loader (individual chains are colored in shades of orange), the DnaB CTD (colored in alternating blue and cyan), and the DnaB NTD tiers (colored in alternating dark and light gray). The DnaB CTD-NTD linker helix (residues 182–202) is colored yellow and the DnaB CTD helix (residues 291–302),which is involved in λP-binding interactions, is colored green. (**D**) Same as panel C except that the model has been rotated by 90°. The five λP molecules in the complex are labeled λP1 (chain Z), λP2 (chain Y), λP3 (chain X), λP4 (chain W), and λP5 (chain V).

DOI: https://doi.org/10.7554/eLife.41140.003

The following figure supplements are available for figure 2:

**Figure supplement 1.** Architecture and biochemistry of the *E.coli* DnaB helicase and the bacteriophage λP helicase loader.
DOI: https://doi.org/10.7554/eLife.41140.005

**Figure supplement 2.** Cryoelectron microscopy and cryoelectron tomography of the DnaB•λP helicase•helicase loader complex.
DOI: https://doi.org/10.7554/eLife.41140.006

**Figure supplement 3.** Quantitative analysis of the DnaB•λP EM density map.
DOI: https://doi.org/10.7554/eLife.41140.007

**Figure supplement 4.** 3D classification of the DnaB•λP EM data.
DOI: https://doi.org/10.7554/eLife.41140.008

**Figure supplement 5.** Interrogation of the DnaB•λP EM data set for additional stoichiometric or conformational states.
DOI: https://doi.org/10.7554/eLife.41140.009

*Figure 2 continued on next page*

*Figure 2 continued*

**Figure supplement 6.** Details of the atomic model and density of the DnaB•λP complex.
DOI: https://doi.org/10.7554/eLife.41140.010

compositional heterogeneity was eliminated when ssDNA derived from the Oriλ phage replication origin was included. Native MS of the BP•origin ssDNA complex revealed a single entity with a mass of 459.5 kDa; this mass corresponds to the $B_6P_5$ complex bound to origin-derived 43-mer ssDNA (*Figure 3D*, *Figure 3—figure supplement 2*, *Table 3* and Appendix). Surprisingly, although ATP was included in both the ssDNA and ssDNA-free samples, native MS measurements showed no evidence that either complex included nucleotide. Nevertheless, orthogonal cryo-EM and MS analyses point to a physiological stoichiometry for the BP complex of $B_6P_5$.

The BP complex presents as a three-layered ensemble with approximate dimensions of 135 Å x 150 Å x 120 Å (*Figure 2*). Two of these layers correspond to the six NTD and CTD components of the DnaB-helicase, and the third layer represents the λP helicase loader. The NTD and CTD layers of the BP complex exhibit a right-handed open spiral configuration, which is distinct from all previously described structures of the DnaB-helicase (*Bailey et al., 2007a*; *Wang et al., 2008*; *Lo et al., 2009*; *Itsathitphaisarn et al., 2012*; *Arias-Palomo et al., 2013*; *Liu et al., 2013*; *Strycharska et al., 2013*; *Bazin et al., 2015*), but is reminiscent of the configuration of DnaB in the 25 Å EM map of *E. coli* DnaB bound to the DnaC helicase (BC) loader (*Arias-Palomo et al., 2013*); a more complete comparison of the two helicase loader complexes must await higher resolution analysis of the BC entity.

The third layer of the BP complex comprises the λP helicase loader (*Figure 2*, *Figure 3—figure supplement 1D*, *Figure 4*). Presence of the λP loader, in combination with the breached interfaces

**Table 1.** Data collection and model refinement.

| Data collection | |
| --- | --- |
| Microscope/Camera | Titan Krios 300kV/Gatan K2 Summit |
| Pixel size (Å) | 1.07 |
| Defocus range (μm) | −1.0 to −3.0 |
| Cell Dimensions | |
| a, b, c (Å) | 273.92, 273.92, 273.92 |
| α, β, γ (degrees °) | 90, 90, 90 |
| *Reconstruction* | |
| Particles | 90,883 |
| Resolution (Å) | 4.1 |
| Model Refinement | |
| Program | Phenix (real_space_refine) |
| Resolution Limit (Å) | 4.1 |
| Number of chains | 11 |
| Number of residues | 3280 |
| RMS bond length (Å) | 0.008 Å |
| RMS bond angle (degrees °) | 1.099 |
| Ramachandran plot | |
| Preferred (%) | 84.56 |
| Allowed (%) | 15.26 |
| Outliers (%) | 0.18 |
| MolProbity | |
| Clash score | 9.87 |
| Rotamer outliers (%) | 0.58 |

DOI: https://doi.org/10.7554/eLife.41140.013

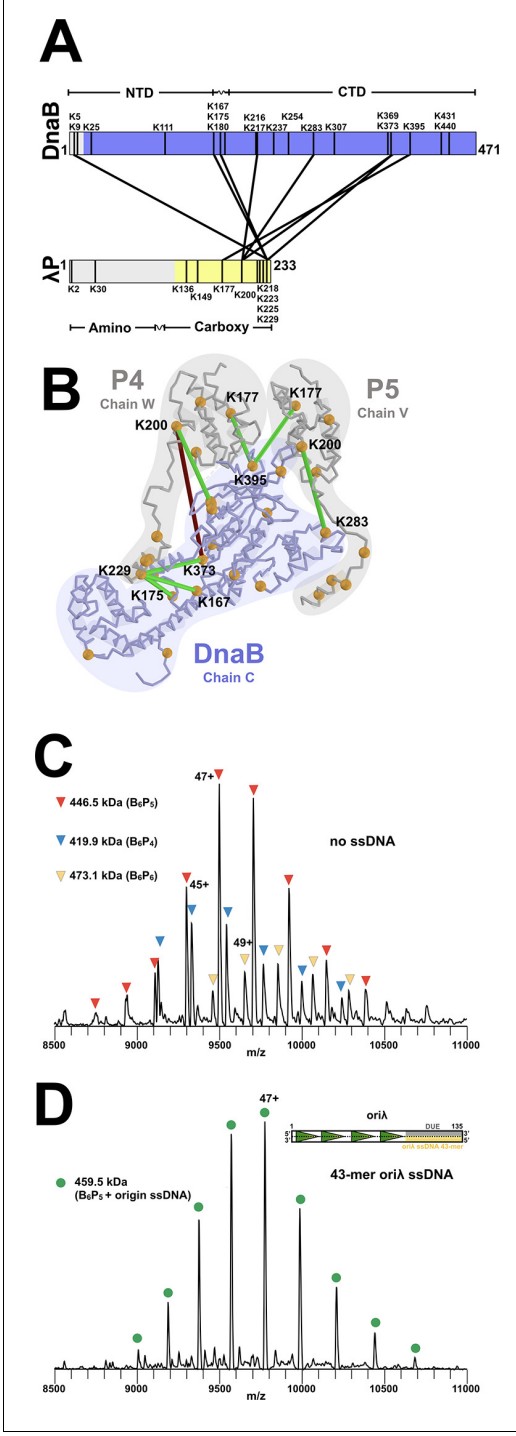

**Figure 3.** Stoichiometry and chain direction in the DnaB• λP complex. (**A**) Linear representations of the amino acid sequences of *E.coli* DnaB and λP colored in purple and yellow, respectively, to indicate the portions of each protein that are visible in our EM density maps. Architectural landmarks of each protein are indicated. The black vertical lines represent lysine residues. The black lines between DnaB and λP represent intermolecular crosslinks provided by the CX-MS *Figure 3 continued on next page*

in the NTD and CTD layers, re-sculpts the central chamber of DnaB into a deep canyon that runs for ~90–100 Å along the entire length of the BP complex; this canyon is essentially open to solvent. The dimensions of the canyon suggest that up to ~30 bp of DNA could be accommodated within; this estimate compares well with measurements on the analogous DnaB•DnaC complex (*Fang et al., 1999*), an entity of similar size to the BP complex.

## The λP helicase loader binds to subunit interfaces of the DnaB-helicase

In the BP complex, five copies of the λP helicase loader bind to five consecutive DnaB subunit interfaces. Each λP protomer is comprised of a helical domain that binds in a deep gulley formed by an interface between two adjacent DnaB CTDs (*Figure 4*). The five λP protomers are deployed in a right handed spiral arrangement and exhibit an average rise per subunit of ~3.1 Å and a pitch of ~10 Å (Below, Materials and methods, Appendix and (*Lu and Olson, 2003*)). Further, each λP protomer sends out a lasso-like segment that runs along the edge of the CTD and positions a λP helix against a site formed by the CTD and the linker helix (residues 183–194) of an adjacent DnaB subunit (*Figure 2C and D*). In addition, owing to intrinsic asymmetry in the DnaB-helicase hexamer (below), disposition of two λP protomers (λP2 and λP4) enables contacts to a nearby NTD (DnaB residues 77–78; λP2 to chain E and λP4 to chain C, *Figure 4B*); this feature gives rise to two types of λP – DnaB interfaces (Appendix) that differ on contacts to the NTD.

Our analysis implies that the BP complex is organized around five sub-structures that each contain a heterotrimer comprised of a single λP and two DnaB chains. The two types of λP•(DnaB)$_2$ interfaces described above each bury a substantial amount of accessible surface area (~2200 Å and ~2600 Å; these values are lower limits owing to our incomplete λP model, which accounts for ~50% of the λP sequence, Appendix). This feature is consistent with the observed high affinity between DnaB and λP (*Mallory et al., 1990*). The finding that λP forms an extensive interface with two flanking DnaB subunits, which comprises a form of molecular handcuffing, leads to the overall conclusion that inhibition of translocation activity could arise through prevention of essential conformational changes by individual CTDs.

*Figure 3 continued*

procedure. Analysis of the BP complex using CX-MS appears in the Appendix section, in **Figure 3—figure supplement 1** and in **Table 2**. (B) The CX-MS derived intermolecular crosslinks are plotted onto DnaB chain C, λP4 chain W, and λP5 chain V of the BP model. All lysine residues are depicted as orange spheres. Lysine residues reported by CX-MS to have been crosslinked are numbered. Lysine residues connected by a line are those which were detected by the CX-MS procedure (**Table 2**). Colored in red is the sole lysine pair whose distance exceeds 30 Å; the other lines capture distances below 30 Å and are colored in green. The various chains of DnaB and λP are outlined. (C) Native MS analysis of the *E. coli* DnaB helicase•bacteriophage λP complex. The most intense peak series corresponds to the $B_6P_5$ assembly, with lower relative intensity peaks indicating presence of subpopulations of $B_6P_4$ and $B_6P_6$. Notably, the mass estimates indicated that no nucleotide was present in any of the three populations of the BP complex, which is distinct from that seen in the native MS of *Aquifex aeolicus* DnaB-DnaC in which five nucleotides were observed (**Figure 8—figure supplement 1**). (D) Incubation of the DnaB•λP sample from panel C with a lambda replication origin-derived 13.1 kDa 43-mer ssDNA yielded a single peak series with a mass corresponding to the $B_6P_5$ assembly with one bound origin-derived ssDNA molecule. As with the samples without ssDNA (panel 3C), the mass estimates indicated that no nucleotide was present in the ssDNA complex. The inset depicts the oriλ replication origin and shows the location of the 43-mer DUE-derived ssDNA (**Learn et al., 1997**) used in this experiment.
DOI: https://doi.org/10.7554/eLife.41140.004

The following figure supplements are available for figure 3:

**Figure supplement 1.** Cross-linking mass spectrometry of the DnaB•λP complex.
DOI: https://doi.org/10.7554/eLife.41140.011

**Figure supplement 2.** Native MS of DnaB•λP with ssDNA.
DOI: https://doi.org/10.7554/eLife.41140.012

## The λP helicase loader directly remodels the CTDs of the DnaB-helicase into an open pseudo-helical configuration

In the BP structure, the CTD layer is found in an open right-handed spiral staircase configuration (**Figure 5A and B** and **Videos 1** and **2**). As oriented in **Figure 5**, chains A and B are at the 'bottom' and 'top' of the staircase, respectively. To gain insights into the transition of DnaB from a closed planar ring (Stage I) to an open right handed pseudo-helical configuration (Stage II) to an closed spiral (Stage IV), we employ the language of helical parameters (Appendix, Materials and methods, and (**Lu and Olson, 2003**)). Our analysis takes note that the configuration of the NTD and CTD tiers of all DnaB structures populate two broad classes of configurations, termed dilated and constricted (**Bailey et al., 2007a**; **Strycharska et al., 2013**). Principally, these forms differ on the width of the internal chamber (constricted:~15 Å, dilated:~50 Å), which reflects distinct organization of the NTD and CTD layers. As well, nucleotide drives transitions between these forms (**Strycharska et al., 2013**). Isolated DnaB (Stage I) is found in two closed planar ring configurations where the NTD and CTD tiers populate either the dilated ((**Bailey et al., 2007a**; **Wang et al., 2008**), PDB: 2R6A) or constricted configuration ((**Strycharska et al., 2013**), PDB: 4NMN). However, the ssDNA bound complex (Stage IV) displays a hybrid arrangement with a constricted CTD layer and a dilated NTD tier ((**Itsathitphaisarn et al., 2012**), PDB: 4ESV). Structural comparisons of various orthologs of DnaB from *E. coli* (Stage II/III) to those from *A. aeolicus* (Stage I (**Strycharska et al., 2013**)) and *G. stearothermophilus* (Stage IV (**Itsathitphaisarn et al., 2012**)) are justified by a high degree of structural and sequence conservation ((**Leipe et al., 2000**), **Figure 5—figure**

supplement 1** and Appendix).

In comparison to the constricted closed planar ring form (PDB: 4NMN), the CTDs in the BP complex exhibit a right-handed pseudo-helical configuration characterized by an average helical rise per subunit value of ~4.1 Å, and a helical pitch of ~16 Å. Furthermore, we find that the CTDs exhibit an average helical twist of ~56.4° along the pseudo-helical axis. By contrast, the CTDs in the ssDNA bound form (Stage IV) exhibit an average helical rise per subunit of ~7.4 Å, a helical pitch of ~27 Å, and an average helical twist of ~60°.

In addition to displacements along the helix axis, the relative inclination of the CTDs is also reconfigured in the loader complex. Superpositions with a $λP_1$•$(CTD)_2$ substructure against all pairs of CTDs from the closed planar ring (constricted and dilated), and the closed spiral forms shows that the CTDs in the BP complex are inclined by ~20° towards the helical axis and the internal chamber, in comparison to those in the dilated closed ring form (**Figure 5C and D**). The resulting reconfiguration of CTDs in the BP complex brings them into an arrangement that is close, but not identical, to

**Table 2.** Crosslinked peptides provided by the CX-MS procedure and their interpretation in terms of the EM model.

| Peptide | Protein 1 | Residue 1 | Protein 2 | Residue 2 | Consistency with BP EM model |
|---|---|---|---|---|---|
| ALAKELNVPVVALSQLNR(4)-ANKDEGPK(3):0 | EcDnaB | 373 | EcDnaB | 175 | Yes |
| VFKIAESR(3)-ANKDEGPK(3):0 | EcDnaB | 167 | EcDnaB | 175 | Yes |
| KTAGLQPSDLIIVAAR(1)-VDQTKIR(5):0 | EcDnaB | 217 | EcDnaB | 283 | Yes |
| ISGTMGILLEKR(11)-VDQTKIR(5):0 | EcDnaB | 307 | EcDnaB | 283 | Yes |
| ALAKELNVPVVALSQLNR(4)-VFKIAESR(3):0 | EcDnaB | 373 | EcDnaB | 167 | Yes |
| ALAKELNVPVVALSQLNR(4)-AGNKPFNK(1):0 | EcDnaB | 373 | EcDnaB | 2 | DnaB residue two is not observed in our map |
| ANKDEGPKNIADVLDATVAR(8)-VFKIAESR(3):0 | EcDnaB | 180 | EcDnaB | 167 | Yes |
| ANKDEGPKNIADVLDATVAR(8)-ALAKELNVPVVALSQLNR(4):0 | EcDnaB | 180 | EcDnaB | 373 | Yes |
| KAADELVHMTAR(1)-ADKRPVNSDLR(3):0 | lambdaP | 177 | EcDnaB | 395 | Yes |
| GEAIPEPVKQLPVMGGR(9)-VDQTKIR(5):0 | lambdaP | 200 | EcDnaB | 283 | Yes |
| INRGEAIPEPVKQLPVMGGR(12)-KTAGLQPSDLIIVAAR(1):0 | lambdaP | 200 | EcDnaB | 217 | Yes |
| ANKDEGPK(3)-FGLKGASV(4):0 | EcDnaB | 175 | lambdaP | 229 | Yes |
| AGNKPFNK(1)-FGLKGASV(4):0 | EcDnaB | 2 | lambdaP | 229 | DnaB residue two is not observed in our map |
| VFKIAESR(3)-FGLKGASV(4):0 | EcDnaB | 167 | lambdaP | 229 | Yes |
| ALAKELNVPVVALSQLNR(4)-GEAIPEPVKQLPVMGGR(9):0 | EcDnaB | 373 | lambdaP | 200 | No |
| ALAKELNVPVVALSQLNR(4)-FGLKGASV(4):0 | EcDnaB | 373 | lambdaP | 229 | Yes |
| AQALAKIAEIK (6)-AKFGLK(2):0 | lambdaP | 218 | lambdaP | 225 | Yes |
| FGLKGASV(4)-IAEIKAK(5):0 | lambdaP | 229 | lambdaP | 223 | Yes |
| GEAIPEPVK QLPVMGGR(9)-KAADELVHMTAR(1):0 | lambdaP | 200 | lambdaP | 177 | Yes |
| IANNMPEQYDEK PQVQQVAQIING VFSQLLATFPASLANR(12)-MKNIAAQMVNFDR(2):0 | lambdaP | 30 | lambdaP | 2 | Lambda P residues 2 and 30 are not observed in our map |

DOI: https://doi.org/10.7554/eLife.41140.014

that required for ATP hydrolysis as inferred from the corresponding nucleotide-binding site in the ssDNA form (*Itsathitphaisarn et al., 2012*).

Changes to the pitch, twist, and inclination of the CTDs during the transition from the closed planar ring form to the open right-handed spiral in the BP complex give rise to an inner chamber in the BP complex with a diameter =~20–25 Å; this value is comparable to the constricted closed spiral form (PDB = 4ESV, diameter =~20–25 Å), and slightly larger than the constricted closed planar form (PDB = 4NMN, diameter =~15 Å), but much smaller than dilated closed planar form (PDB = 2R6A, diameter =~50 Å). Thus, the CTD tier in the BP complex adopts the constricted configuration.

The λP helicase loader reconfigures the CTD layer in the BP complex into a conformation that differs substantially from previously described DnaB structures. Reconfiguration arises from three broad types of changes: (1) displacements of the CTDs along the pseudo-helical axis to form the open right-handed configuration; the average pitch of the resulting entity is ~40% shorter than the ensemble in the ssDNA bound complex, (2) changes to the helical twist of individual CTDs along the helical axis; the CTDs in the BP complex are underwound by ~3.6° with respect to those in the closed planar ring and closed spiral forms, and (3) inclination of the CTDs toward the helical axis; this change yields nucleotide-binding sites with configurations that are nearly optimal for catalysis (below). Collectively, these changes cause the rupture of one of the six CTD interfaces, between subunits A and B, and produce a ~ 15 Å gap between the two CTDs that span the breached interface; this gap is large enough to permit entry of ssDNA into the central chamber of the helicase. These findings have significant implications for opening and closing of the DnaB helicase during recruitment to origin DNA.

**Table 3.** Mass measurements from the native MS analysis of DnaB and λP assemblies.

| Sample condition | Measured Mass ± SD (Da)[*] | | | Assemblies | Expected mass (Da)[†] | Δ mass (Da) | % Mass Error |
|---|---|---|---|---|---|---|---|
| BP sample in 450 mM ammonium acetate, pH 7.5, 0.5 mM magnesium acetate, 0.01% Tween-20 | | | | | | | |
| | 4,46,500 | ± | 60 | $B_6P_5$ | 4,46,145 | 355 | 0.08 |
| | 4,73,100 | ± | 50 | $B_6P_6$ | 4,72,663 | 437 | 0.09 |
| | 4,19,950 | ± | 50 | $B_6P_4$ | 4,19,627 | 324 | 0.08 |
| BP sample + oriλP ssDNA in 450 mM ammonium acetate, pH 7.5, 0.5 mM magnesium acetate, 0.01% Tween-20 | | | | | | | |
| | 4,59,480 | ± | 15 | $B_6P_5$ + one oriλP ssDNA | 4,59,285 | 195 | 0.04 |
| BP sample in 500 mM ammonium acetate, 0.01% Tween-20[‡] | | | | | | | |
| | 4,46,270 | ± | 20 | $B_6P_5$ | 4,46,145 | 125 | 0.03 |
| | 4,72,840 | ± | 20 | $B_6P_6$ | 4,72,663 | 177 | 0.04 |
| | 4,19,750 | ± | 15 | $B_6P_4$ | 4,19,627 | 124 | 0.03 |

[*] Calculated from the average and corresponding standard deviation of all the measured masses across the charge-state distribution ($n \geq 4$). Only the peak series with signals above 10% relative intensity were processed and deconvoluted.

[†] The expected masses include DnaB (N-terminal Met loss), 52,259 Da; λP, 26,518 Da; Oriλ-derived ssDNA (5' and 3'-OH), 13,141 Da.

[‡] Better mass accuracies were observed for protein samples in ammonium acetate without magnesium acetate due to the absence of magnesium adduction.

DOI: https://doi.org/10.7554/eLife.41140.015

## The DNA and ATP-binding sites are disrupted in the DnaB• λP complex

Prior studies have established that, within the BP complex, ssDNA binding is altered with λP making most, if not all, of the contacts (*Learn et al., 1997*). In addition, ATPase, and concomitantly, the helicase activities of the DnaB-helicase are suppressed by λP (*Mallory et al., 1990*). However, the λP-binding site is more than 10 Å from both the ATP and DNA-binding sites. Suppression of DNA binding and ATP hydrolytic activities must, therefore, arise indirectly, as a consequence of the structural changes described above. We find that alterations in the helical pitch, twist, and inclination of the CTD pseudo-helix in the BP complex combine to critically distort positions of six DNA-binding loops and the disposition of the subunits that form the composite ATP-binding sites. By example, in the ssDNA complex, side chains from the DNA-binding loops (*G.st*: R381, E382, G384; *E. coli*: R403, E404, G406) of each DnaB protomer contact two phosphate groups per subunit along the helical path of the CTDs (*Itsathitphaisarn et al., 2012*). However, in the BP complex, distortions in the disposition of the CTDs significantly shifts the corresponding side chains by distances ranging from ~3 to ~ 18 Å (*Figure 6A*).

The DnaB-helicase specifies six recA style nucleotide-binding sites, each of which resides at a protomer interface. Although our EM samples were prepared with a large excess of ATP, we observe that five of the six sites in the BP complex are occupied by ADP, while the sixth site, whose composite architecture involves subunits that span the breached CTD ring, lies unfilled (*Figure 2—figure supplement 6*). Moreover, alterations to the relative orientations of the CTDs in the BP complex have remodeled the five filled sites into a configuration that is likely not optimal for catalysis. In the ssDNA complex, the Walker A and Walker B motifs from one CTD partner with K418 and R420 from the adjacent CTD to assemble an ATP-binding site (Appendix). In the BP complex, however, we observe changes in the relative disposition of the CTDs that shift the positions of the alpha carbons associated with K440 (homolog of *B.st* K418 (the ssDNA complex)) and R442 (homolog of *B.st* K420) away from those seen in the translocating ssDNA complex (*Figure 6B*). For some of the sites, K440 and R442 are resolved in the density maps; these also appear to be shifted in comparison to the ssDNA complex (*Figure 6B* and *Figure 2—figure supplement 6B*). We ascribe the sub-optimal arrangement of catalytic sites to presence of the λP loader, however, we cannot exclude that absence of ATP prevents optimal alignment. Additionally, the small structural changes reported here should be taken as tentative owing to the resolution of our maps.

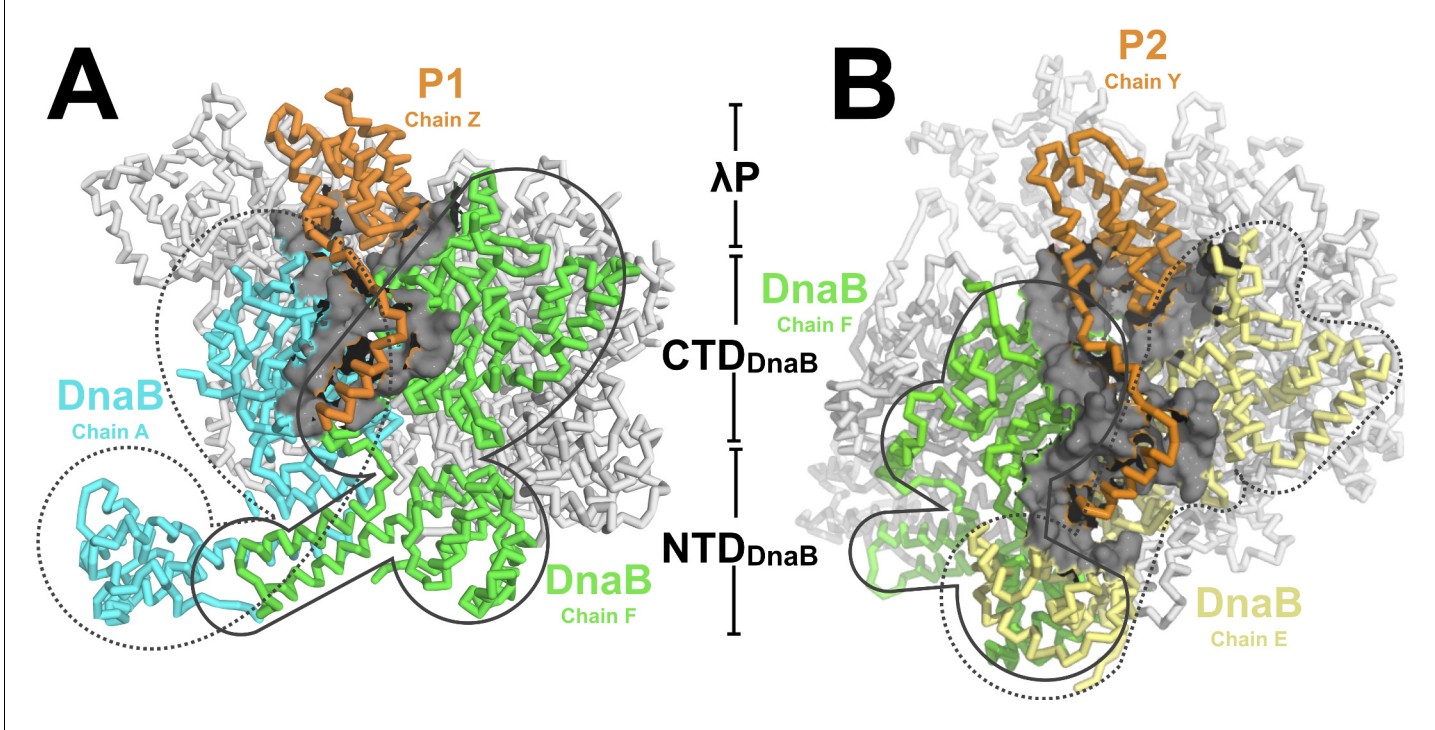

**Figure 4.** The DnaB•λP complex features two types of interfaces. (**A**) An example of the first type of interface in the BP complex occurs between λP1 (chain Z, orange) and chain A (cyan) and chain F (light green) of DnaB. Other DnaB chains are colored in white. The surface representation (gray) includes DnaB residues within 10 Å of an alpha carbon from λP1 (chain A: residues 232, 278–306, 387–395, 419–432, 454–457; chain F:182–203, 213, 214, 217, 388–403,419-421, 429–439, 445–450, 463–468). This interface includes only contacts to the DnaB CTDs of the above chains. The BP complex includes three instances of this type of interface (to λP protomers: λP1, chain Z; λP3, chain X; and λP5, chain V). (**B**) A second type of interface in the complex is comprised of λP2 (chain Y, orange) and chain F (light green) and chain E of DnaB (light yellow). Other DnaB chains are colored in white. The surface representation (gray) includes positions in either of the above chains of DnaB that come within 10 Å of λP2. In this second type of interface, λP makes contacts to the CTD and NTD of the above DnaB chains. The BP complex includes two instances of this type of interface (to λP protomers: λP2, chain Y; and λP4, chain W).

DOI: https://doi.org/10.7554/eLife.41140.016

Taken together, our findings indicate that the λP helicase loader induces a conformation of the DnaB helicase that is neither optimal for DNA binding nor ATP hydrolysis. Observation of a distorted ssDNA-binding site implies that the DnaB-helicase, while complexed to λP, may not bind ssDNA as in the Stage IV translocating complex. This result is consistent with suppression of a crosslink between DnaB and ssDNA derived from the Oriλ replication origin when the λP loader is present (*Learn et al., 1997*). The λP-enforced misalignment of the composite ATP-binding sites may also explain lack of ATPase activity in the BP complex.

### The λP helicase loader allosterically cracks an interface in the NTD layer

The λP helicase loader makes few contacts to the NTD tier of DnaB in the BP complex, nevertheless, this tier is also reconfigured (*Figure 7* and *Videos 2* and *3*). The likely driver of rearrangement is the λP-enforced remodeling of the CTD tier, which provides a reconfigured surface against which the NTD layer must pack; contacts by the λP2 and λP4 protomers to the NTD tier may also contribute, but to a small extent (*Figure 4B*). In comparison to the Stage I closed planar ring, the NTD layer in the BP complex exhibits an open spiral arrangement. As with every other DnaB structure, the six N-terminal domains of the NTD layer feature a trimer of dimers configuration that displays pseudo-three-fold symmetry. The dimers that comprise the NTD layer in the BP complex are closely related to those in other DnaB structures (RMSD: 1.3 Å, *Figure 5—figure supplement 1D,E* and Appendix). However, the arrangement of dimers in the complete NTD layer exhibits a conformation that is distinct from that of other DnaB structures (*Figure 7*), though it is reminiscent to that seen in the DnaB

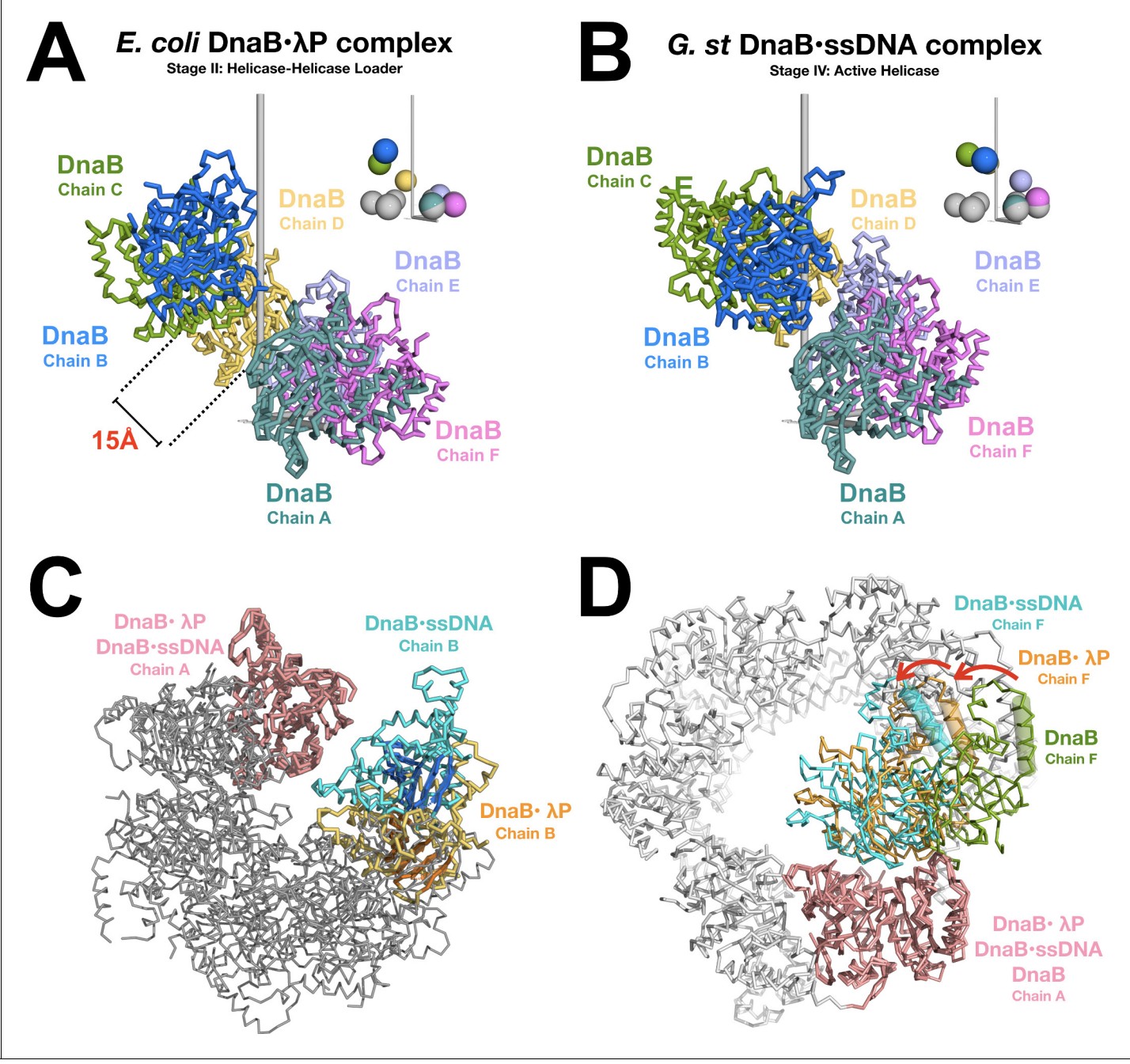

**Figure 5.** Opening and closing of the CTD tier of the DnaB helicase. (**A**) The CTD layer in the DnaB helicase in the BP complex adopts a right handed pseudo-helical configuration after the transition from the closed planar (PDB = 4NMN) to the open spiral form of the BP complex. The CTD layer exhibits a pitch of ~16 Å. DnaB from the complex is depicted in a ribbon representation, colored by subunit and labeled by chain. The inset shows colored spheres drawn at the center of mass of each CTD using the same color as the associated subunit. The gray spheres in the inset represent centers of mass for CTDs from the closed planar constricted form of DnaB (PDB = 4NMN). The pseudo-helical axis is aligned with the vertical axis, which is shown in gray. (**B**) Same as panel A except that the CTD layer from the DnaB helicase in the ssDNA complex (PDB = 4ESV) is shown. The 4ESV CTD layer adopts a pitch of ~27 Å. DnaB from the ssDNA complex is depicted in a ribbon representation, colored by subunit and labeled by chain. As in panel A, the inset shows colored and gray spheres drawn at the centers of mass of each CTD from DnaB in the ssDNA complex and the closed planar constricted form (PDB = 4NMN), respectively. (**C**) Closing of the CTD tier is inferred from comparison of the open spiral in the BP complex to the closed spiral form of the ssDNA complex (PDB = 4ESV). The DnaB helicase from the BP complex and the ssDNA complex are superimposed on subunit A (both colored pink). Subunit B, which lines the ruptured interface in the BP complex is colored orange. The beta sheets of the RecA style fold are shown in a cartoon representation. The corresponding subunit in the closed spiral ssDNA complex is colored cyan. The remaining DnaB subunits are colored gray. (**D**) Inclination of individual subunits of DnaB towards the helical axis during opening and closing of the helicase. The open spiral BP

*Figure 5 continued on next page*

*Figure 5 continued*
complex (orange), the closed spiral ssDNA complex (blue, PDB = 4ESV), and the closed planar dilated form (green, PDB = 2R6A) are superimposed on subunit A of the DnaB helicase (pink). Changes in the relative orientation of subunits were calculated by measuring the degree of rotation of the next subunit (chain F) in the helicase. One structurally conserved CTD helix (residues 293–305 of BP, residues 272–284 of the closed spiral ssDNA complex and residues 271–285 of the closed planar dilated form) is indicated by a cylinder. Other subunits in the closed planar dilated form (PDB = 2R6A) structure are colored white.
DOI: https://doi.org/10.7554/eLife.41140.017
The following figure supplement is available for figure 5:

**Figure supplement 1.** Conservation of the amino acid sequence and structure of the DnaB helicase.
DOI: https://doi.org/10.7554/eLife.41140.018

segment of the low-resolution EM map of the *E. coli* DnaB•DnaC helicase loader complex (*Figure 8—figure supplement 1B* and Appendix).

The NTD layer from the BP complex features a central chamber in the constricted state, surrounded by component domains in an open spiral configuration. The diameter of the BP NTD layer (~20–25 Å) is similar to that seen in the closed constricted planar form, and differs markedly from the ~50 Å diameter of the corresponding structure in the ssDNA complex, which adopts the dilated conformation. Comparison of the helical parameters of the NTD tier from the BP complex (Materials and methods) to those of the constricted closed ring form revealed an average helical rise per subunit of ~2.6 Å, a helical pitch of ~7 Å, and helical twist values per subunit that ranged from ~40° to ~70° along the pseudo-helical axis. Direct comparison between the open NTD spirals in the BP and ssDNA complex is complicated by their distinct configurations, constricted and dilated. Nevertheless, we find that the ssDNA complex shows similar pseudo-helical parameters: helical rise per subunit value of ~2.6 Å, a helical pitch of ~7.6 Å, and helical twist values per subunit that ranged from ~45 to 71° (comparison of PDB entries: 4ESV and 2R6A). Reconfiguration of the NTD layer in the BP complex into an open spiral creates a ~ 20 Å gap between the disrupted subunit interface. As with the opening in the CTD layer, the breach in the NTD layer is sufficiently large to provide access to the central chamber of DnaB to ssDNA.

Direct remodeling of the CTD tier by the λP helicase not only allosterically remodels the NTD tier, but also changes the relationship between the NTD and CTD tiers in the complete DnaB helicase. In the closed planar structure, these two layers are essentially parallel. By contrast, the CTD and NTD layers of DnaB in the BP complex make an angle of ~15°; the corresponding value for the Stage IV form is ~7°. The change in the relationship between the two layers is also seen in the surface area buried. The NTD and CTD layers of the closed planar ring bury an extensive surface area (~7100 Å$^2$). By contrast, the corresponding value for the BP complex is ~3100 Å$^2$; this implies that the NTD and CTD layers are held considerably less tightly in the loader complex than in the closed ring.

We find that even in the absence of an extensive interface with the NTD layer, the λP helicase loader influences its organization. Reconfiguration of the CTD layer by λP programs the open spiral configuration of the NTD layer, as it also alters the relationship between the layers. We also observe that alteration in the diameter of the central chamber is one change that accompanies transition of DnaB from the loader bound complex to that in the translocation state (*Figure 7*). Collectively, analysis of the allosteric

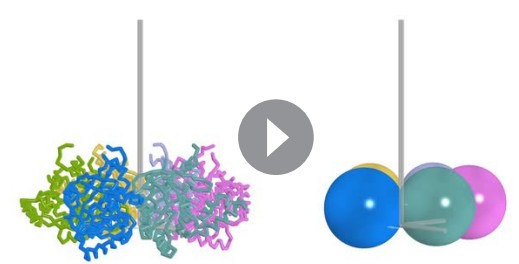

**Video 1.** Opening the CTD layer of the DnaB-helicase by the λP helicase loader. This movie depicts the transition of the CTD layer of the DnaB helicase from the closed planar constricted form (Stage I, PDB = 4NMN) to the right handed open spiral form (Stage II, this work). The CTDs are depicted in a ribbon representation (left) and as spheres drawn around the respective centers of mass (right). The ribbon and sphere representations are colored as in *Figure 5A*. The pseudo-helical axis is aligned with the vertical. In *Videos 1* and *3*, only the end states arise from experimentally determined coordinates; the intermediate structures were calculated by the morph algorithm (PyMOL), and, as such, may be incomplete or contain errors.
DOI: https://doi.org/10.7554/eLife.41140.019

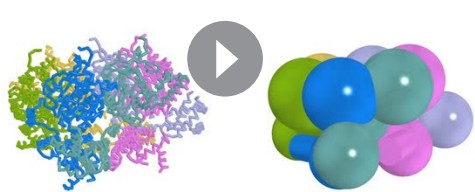

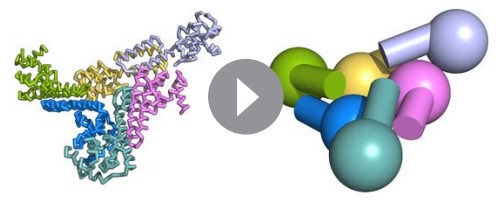

**Video 2.** Opening and closing of the DnaB-helicase. This video depicts the transition of the CTD and NTD layers from the DnaB helicase from the closed planar form (Stage I, PDB = 4NMN) to the right-handed open spiral form (Stage II, this work) to the right- handed closed spiral form of Stage IV (PDB = 4ESV). The DnaB protomers are depicted in a ribbon representation (left) and as spheres drawn around the respective centers of mass of each CTD and cylinders/spheres drawn around the around the respective centers of mass of the globe domain and helical hairpin of each NTD (right). Five molecules of the λP helicase loader (depicted in a ribbon representation and colored in gray) bind to the DnaB helicase and mediate the transitions depicted in the video. ssDNA (depicted in a sticks representation and colored in red) is depicted as binding to the helicase•loader complex prior to transition to the Stage IV conformer. The DnaB protomers are colored as in *Figure 5A*. For *Video 2*, only the end and mid states arise from experimentally determined coordinates; the intermediate structures were calculated by the morph algorithm (PyMOL), and, as such, may be incomplete or contain errors.

DOI: https://doi.org/10.7554/eLife.41140.023

**Video 3.** Opening of the NTD layer of the DnaB-helicase by the λP helicase loader. This movie depicts transition of the NTD layer of the DnaB helicase from the closed planar constricted form (Stage I, PDB = 4NMN) to the right-handed open spiral form (Stage II, this work). The NTDs are depicted in a ribbon representation (left) and as cylinders/spheres drawn around the respective centers of mass of the globe domain and the helical hairpin of each NTD (right). The ribbon and cylinder/sphere representations are colored as in *Figure 5A*.

DOI: https://doi.org/10.7554/eLife.41140.022

reconfiguration of NTD layer in the helicase loader complex and comparisons to the helicase entity in the ssDNA complex have significant implications for opening and closing of the DnaB helicase (*Figure 8—figure supplement 2* and *Videos 2* and *3*).

## Discussion

Assembly of the replicative helicase at the appropriate time and place on the genome is an important decision taken by all cells on the path to cell division. In bacteria, the replicative helicase is a closed protein ring, which requires loading factors for assembly and activation on ssDNA (*Bell and Kaguni, 2013*; *Bleichert et al., 2017*). In archaea and eukaryotes, the replicative helicase appears to exist in open and closed states, and the mechanisms of opening and closing are more complex (*Abid Ali and Costa, 2016*; *Parker et al., 2017*; *Zhai and Tye, 2017*; *Li and O'Donnell, 2018*). The present work, in combination with structural and biochemical analyses of other stages of the helicase assembly pathway, provides unprecedented insights into mechanisms of opening, entry of ssDNA into the internal chamber, and closing of the hexameric DnaB-helicase (*Figure 8* – and *Figure 8—figure supplement 2*, Appendix and *Videos 1*, *2* and *3*).

### Mechanism of opening the DnaB-helicase

The isolated *E. coli* DnaB helicase (Stage I) is known to be in equilibrium between two forms: dilated and constricted, with the dilated form representing the ground state; notably, nucleotide shifts the equilibrium towards the constricted state (*Strycharska et al., 2013*). However, binding of the λP helicase loader is not compatible with the dilated form; indeed, the NTD and CTD tiers of the BP complex, by virtue of arrangement and size of internal chamber, are found in the constricted form. As such, we envision that, during transition to Stage II of the assembly pathway, the ring-breaking λP helicase loader opens the closed DnaB-helicase by binding to the constricted form and forcing the CTD layer into an open right-handed pseudo-helical configuration (*Figures 5* and *8*, and *Videos 1* and *2*). Opening is accompanied by changes in the inclination and twist of individual CTDs with respect to the pseudo-helical axis. As well, reconfiguration leads to breach of one DnaB interface and a ~ 15 Å opening in the CTD layer. Changes to the CTD layer alter the relative disposition

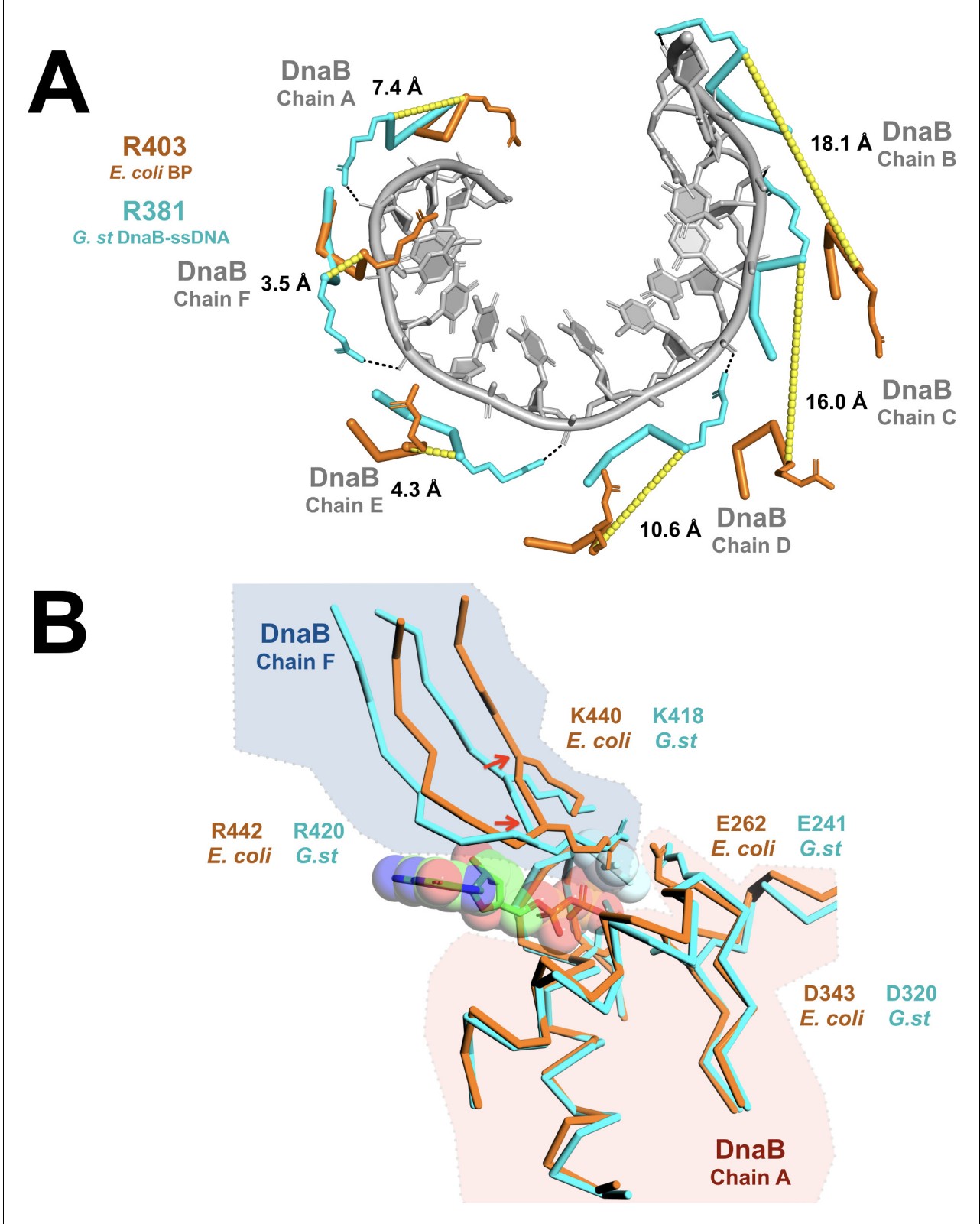

**Figure 6.** Remodeling of the CTD tier shifts the ssDNA binding and ATP hydrolytic sites into inactive configurations. (A) The effect of remodeling of the CTD tier by λP is seen in the misalignment of the DNA-binding residues in the BP complex when compared to the ssDNA complex. In the ssDNA complex, three side chains (*G.st*: R381, E382, G384; *E. coli*: R403, E404, G406) from each DnaB protomer contact two consecutive phosphate groups; for clarity, only R381/R403 are shown. In the BP complex, the altered positions of these residues distorts the binding site, which may preclude interactions

*Figure 6 continued on next page*

*Figure 6 continued*

with ssDNA. The DnaB portions of the BP and ssDNA complexes were superimposed on chain A of DnaB. The ssDNA component from the translocating DnaB helicase structure (PDB = 4ESV) is depicted as a gray cartoon, and amino acids that contact the phosphate backbone are shown as sticks and colored in cyan (*G.st*, ssDNA complex) or orange (*E. coli*, BP complex). Distances between the α-carbon of the corresponding residues in the BP and ssDNA bound complexes are indicated and marked with a yellow dashed line. The approximate position of each chain of DnaB is also indicated. (**B**) Superposition of the nucleotide-binding sites from the open spiral (this work, colored orange) and the closed spiral ssDNA complex (PDB = 4ESV, colored cyan). The catalytic glutamate (E262) of the BP complex is sub-optimally positioned for hydrolysis, as are the α-carbons of the two nucleotide- binding residues: K440 and R442 (indicated with red arrows). The GDP and aluminum fluoride from the closed spiral ssDNA complex are depicted in a ball and stick representation, with transparent spheres. Amino acid residues in DnaB implicated in hydrolysis are shown in a ball and stick representation. Other parts of the DnaB helicase are depicted in ribbon representation. A complete presentation of the nucleotide-binding sites appears in *Figure 2—figure supplement 6*.

DOI: https://doi.org/10.7554/eLife.41140.020

of the NTD and CTD layers, and specify remodeling of the NTD layer into an open spiral. As with the CTD layer, one interface in the NTD tier is ruptured to produce a ~ 20 Å opening in this layer (*Figures 7* and *8*, and *Videos 2* and *3*). Thus, binding of the λP helicase loader produces openings in the DnaB-helicase that are of sufficient size to permit access of ssDNA to the inner chamber.

It is, however, striking that the helicase is not just opened by the λP loader. Rather, it is forced into a configuration that represents an intermediate between the closed planar ring of the isolated helicase and the closed spiral of the ssDNA complex, albeit much closer to the latter than the former. We speculate that the loader forces the helicase into a tense or high-energy configuration, akin to a spring-loaded mouse-trap. This state prepares the helicase to accept ssDNA into its inner chamber, but prevents productive ATP hydrolysis or translocation on ssDNA via the mispositioning of critical amino acids, as suggested by biochemical studies (*Mallory et al., 1990*). Thus, the loader may impede translocation by handcuffing the DnaB CTDs and the CTD-NTD linkers to prevent essential conformational changes expected of the 'hand over hand' mechanism of the translocating species (*Itsathitphaisarn et al., 2012*). Configurational proximity of DnaB in the BP complex to the active conformation poises this state for translocation, requiring only the activating signal. The precise role for nucleotide binding and hydrolysis in the BP complex requires further clarification since the EM maps show presence of 5 ADP molecules (the site that spans the breach is empty), however, no nucleotide is seen in the native MS mass measurements.

An overall consequence of the formation of the BP complex is a change in the ssDNA-binding profile from non-specific in the isolated DnaB helicase to a preference for a sequence from the melted Oriλ origin in the BP complex (*Figure 3D*, *Figure 3—figure supplement 2, and Appendix*). This finding sheds new light on Stages II and III of the assembly pathway. We suggest that the ssDNA-binding site on DnaB in the BP complex is severely compromised, if not completely disrupted, and unable to form the non-specific contacts to the phosphate backbone seen in the translocating form. Furthermore, the specific ssDNA-binding activity that we observe in the BP complex is likely distinct from that in the translocating form, and, we suggest, emerges from contacts made by the λP loader and not DnaB; these findings were anticipated by prior studies (*Learn et al., 1997*; *Shao, 2006*). During Stages II/III, the specific contacts to origin ssDNA may not only act as a handbrake to oppose motion of the helicase, but also hint at a previously undescribed activity by λP as an origin specificity factor, which mediates delivery of DnaB to a specific sequence at the origin.

We observe that opening the DnaB-helicase requires five λP helicase loaders, which bind to consecutive interfaces; the ruptured sixth interface is not able to bind a sixth loader. The peculiar arrangement of isomers of DnaB protomers in the closed ring form suggests that the closed ring form of DnaB harbors two types of NTD interfaces (the CTD interfaces are equivalent), one of which (tail-to-tail) will likely require more energy to open than the second (head-to-tail) ((*Wang et al., 2008*; *Strycharska et al., 2013*), *Figure 7*, and Appendix). Indeed, the interface that is breached in the BP complex is the weaker head-to-tail interface.

The distance between λP protomers does not suggest an extensive interface. Thus, we speculate that binding of individual loaders to DnaB could take place stochastically, with each binding event remodeling an interface until five have bound. In the case where the sixth interface is of the weaker head-to-tail type, the helicase ring would be opened as in the present structure. We note that nothing prevents assembly of an alternate B₆P₅ complex in which the NTD tier interface to be disrupted

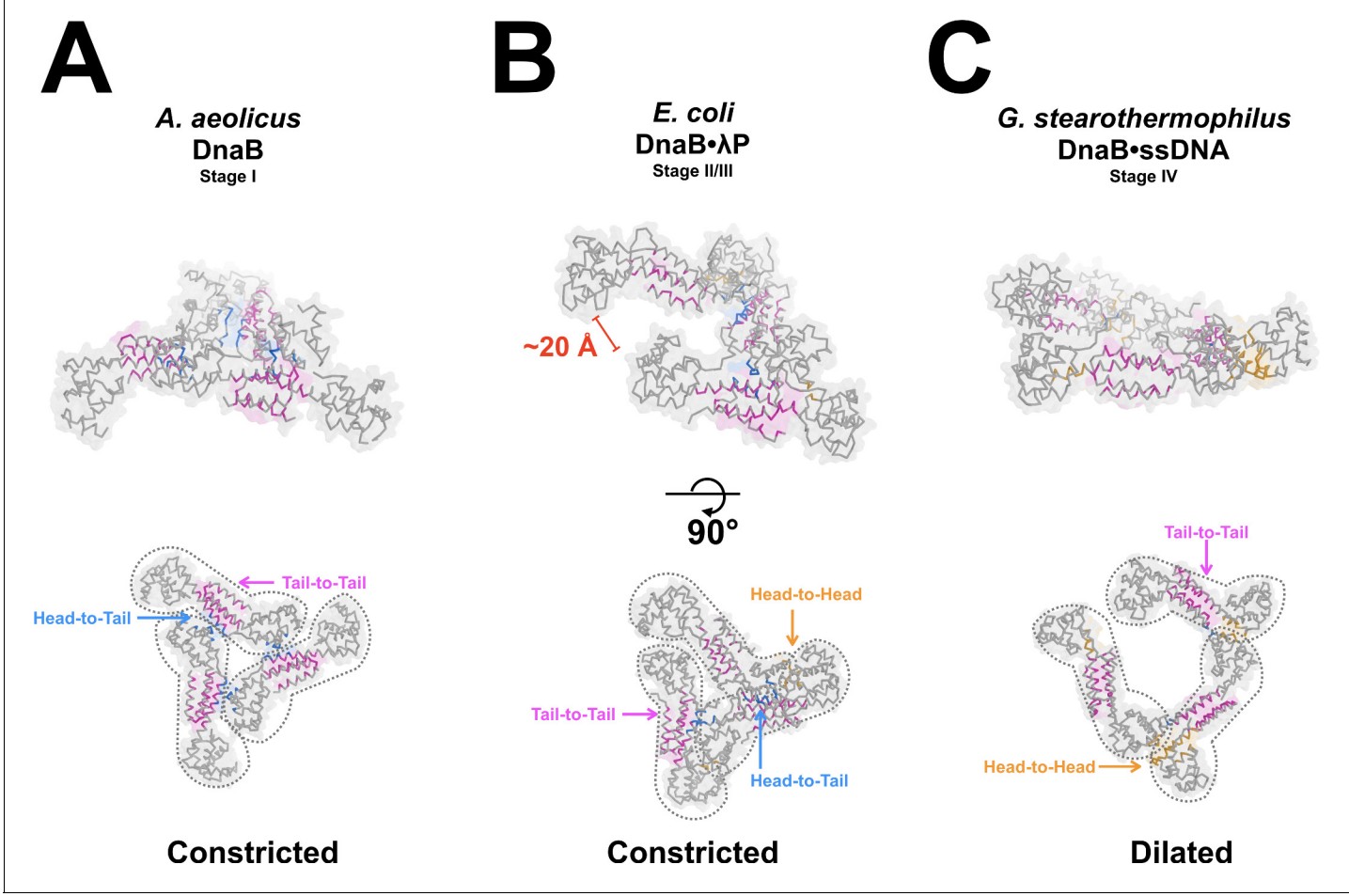

**Figure 7.** Remodeling of the NTD tier of the DnaB helicase by the λP helicase loader. The NTD layers from the various stages of the helicase assembly pathway are compared. Three types of contacts between the NTDs of DnaB monomers have been described among the various forms of DnaB: tail-to-tail (colored in magenta), head-to-tail (colored in blue), and head-to-head (colored in orange). Residues within 4 Å across a particular interface are colored. Head refers to the globular domain in the NTD (*A. aeolicus*: 8–109, *E. coli*: 32–123, *G. stearothermophilus*: 1–112) and tail refers to a pair of helices that comprise a helical hairpin in the NTD (*A. aeolicus*: 110–149, *E. coli*: 124–173, *G. stearothermophilus*: 113–151). The top set of images is rotated by 90° relative to the bottom images. (A)The NTD layer (PDB = 4NMN) from the constricted closed planar configuration in Stage I of the helicase assembly pathway. (B)The NTD layer from the BP complex. Binding of the λP helicase loader to DnaB reconfigures the NTD layer into an open spiral; reconfiguration breaches one of the head-to-head interfaces to create a ~ 20 Å opening. The NTD layer from the BP complex adopts the constricted configuration as defined by the width of the central chamber. (C)The NTD layer from the ssDNA bound form of DnaB (PDB = 4ESV). This NTD layer adopts an open spiral configuration wherein one head-to-head interface is disrupted. Unlike the BP the complex, however, the central chamber in the ssDNA complex is topologically closed through interactions between the NTD and CTD tiers (not shown). The NTD layer is found in the dilated configuration.

DOI: https://doi.org/10.7554/eLife.41140.021

would be of the tail-to-tail type; we speculate that such a complex may not open, and, thus, be non-productive for loading onto DNA.

## Mechanism of closing of the DnaB-helicase

Activation of the DnaB-helicase for translocation on ssDNA requires eviction of the ring-breaking helicase loader and concomitant sealing of the breaches in the NTD and CTD layers (*Figure 8—figure supplement 2* and *Video 2*). An extensive body of insightful work has documented engagement of the bacterial DnaK/DnaJ/GrpE chaperone apparatus at the lambda origin to partially dissolve the λP helicase loader from the BP complex (*LeBowitz et al., 1985*; *Mensa-Wilmot et al., 1989*; *Wyman et al., 1993*; *Polissi et al., 1995*). Comparisons to the ssDNA-DnaB complex (Stage IV)

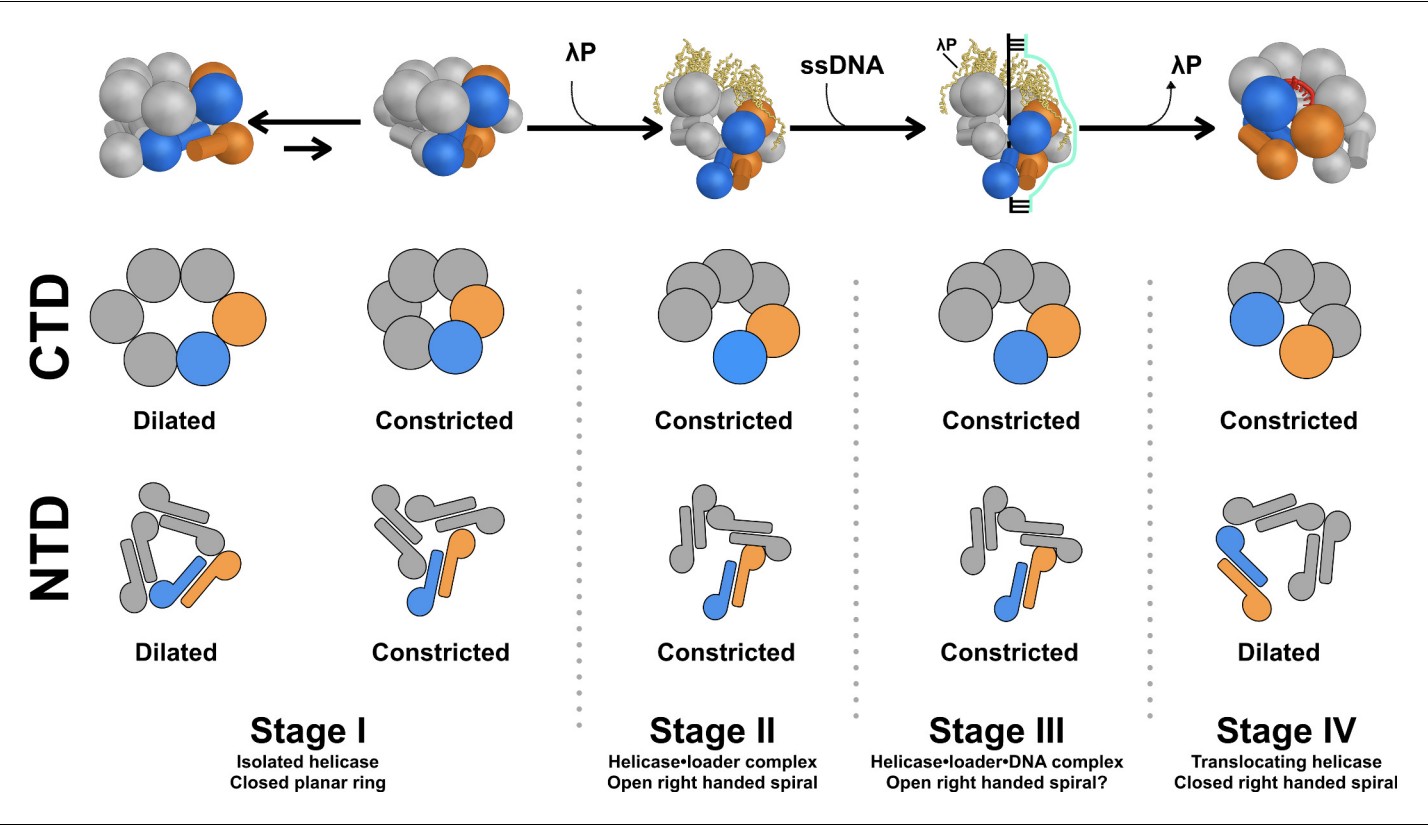

**Figure 8.** Model for assembly of the bacterial DnaB replicative helicase at the origin of DNA replication. The overall structure and conformational state of DnaB in the four stages of the assembly pathway are schematically depicted to include insights from this work. The structure of DnaB is depicted in cartoon form with spheres drawn in place of each CTD and a sphere/cylinder in place of each NTD globe/helical hairpin tail. Each subunit is colored gray, except for the Stages I, II, and III chains A and F, which are shown in blue and orange, respectively. In the Stage IV model, chains A and B are colored orange and blue, respectively. The five copies of the λP helicase loader are depicted as yellow ribbons. A two dimensional representation of the configuration of each tier in the various stages is drawn below each cartoon, colored in the same way. In Stage I of the assembly pathway, DnaB is found in equilibrium between two conformers, termed dilated and constricted. This distinction applies to both the NTD and CTD layers. Addition of the λP helicase loader results in a conformational change in DnaB in which the CTD and NTD tiers are ruptured (Stage II); in this stage, both the CTD and NTD tiers adopt the constricted configuration (this work). In Stage III, the helicase•helicase loader complex engages ssDNA and the initiator protein (not shown) at the oriλ replication origin (produced by prior action of the initiator protein, not shown). Little is known about the path of ssDNA through the complex, as such, it is modeled as a simple cartoon. It is anticipated that both the NTD and CTD tiers in Stage III will retain the constricted configuration of Stage II. Expulsion of the loader protein accompanies transition to Stage IV of the pathway. The CTD layer closes and remains in the constricted conformation. The NTD layer is also sealed, however, its configuration assumes the dilated conformer.
DOI: https://doi.org/10.7554/eLife.41140.024

The following figure supplements are available for figure 8:

**Figure supplement 1.** Comparison of bacterial helicase•helicase loader complexes.
DOI: https://doi.org/10.7554/eLife.41140.025

**Figure supplement 2.** Models for closing of the DnaB helicase after expulsion of the λP helicase loader.
DOI: https://doi.org/10.7554/eLife.41140.026

allow us to infer closing mechanisms that will accompany eviction of the λP loader during transition of the helicase to the active entity. In the first instance, eviction of the loader from DnaB allows the helical CTD tier to relax into the configuration of the ssDNA complex by deploying to its full helical pitch (~27 Å). Second, departure of the loader permits the CTDs to adopt a relaxed ~60° helical twist relationship around the pseudo-helical axis. Third, the DNA binding and the ATPase sites, freed of constraints enforced by λP, assume productive conformations. Origin-derived ssDNA enters the inner chamber with expulsion of the loader. One might speculate that ssDNA in the inner chamber could drive closure of the CTD layer forward in the assembly pathway to the spiral form of the ssDNA complex, rather than backwards to the planar form. Such a mechanism could also ensure

quality control in that non-productive BP complexes without ssDNA do not mature into the active DnaB conformation. We speculate that expulsion of λP enables exchange of ADP in the BP complex for ATP, which must take place for DnaB to assume the translocation competent form (*Figure 8—figure supplement 2*). As well, the unfilled nucleotide site at the breached interface will bind ATP, and this will also further promote closing of the CTD tier. Lastly, the CTD tier, freed of restraining interactions imposed by the λP loader, can undergo conformational changes expected during translocation (*Itsathitphaisarn et al., 2012*).

Relaxation of the CTD tier owing to eviction of the helicase loader provides a reconfigured surface against which the NTD layer is remodeled. However, in contrast to the CTD layer, inferences about mechanisms of closing of the NTD layer depend on precise mapping of DnaB subunits in the spirals seen in Stages II/III and IV of the helicase loading pathway. In both entities, the spiral staircase-like configuration of the DnaB helicase can be considered to have distinct subunits populating its 'top' (Chain B) and 'bottom' (Chain A) ('top' and 'bottom' of both spirals are separately defined based on positions of the CTD domains in the BP and ssDNA complexes). In addition, for our purposes here, 'top' and 'bottom' are inverted from that in (*Itsathitphaisarn et al., 2012*). We note that there are two major ways to map subunits between DnaB spirals in Stages II/III and in Stage IV; these mappings imply radically different mechanisms of helicase closing, especially with respect to the NTD tier (*Figure 8—figure supplement 2*). In closing Scheme I, the 'top' and 'bottom' subunits of each DnaB spiral are mapped as follows: BP: chain A → ssDNA: complex chain A, BP: chain B → ssDNA complex: chain B, etc. (We note that mapping in this context does not imply a structural alignment, but rather a point of spatial orientation). Notably, this arrangement maps the breached CTD interfaces, and concomitantly, the unfilled nucleotide-binding site, from both forms of DnaB onto one another. Also, all hexameric DnaB structures feature an arrangement of alternately configured monomers in which the orientation of the NTD is the locus of the differing conformations ((*Bailey et al., 2007a*; *Wang et al., 2008*; *Lo et al., 2009*; *Itsathitphaisarn et al., 2012*; *Stelter et al., 2012*; *Liu et al., 2013*; *Strycharska et al., 2013*) and *Figure 7*). Termed *cis* (inner) and *trans* (outer), these conformations describe distinct configurations of DnaB monomers. In the *cis* monomer, a helical hairpin substructure in the NTD points toward the CTD of the same chain, while in the *trans* monomer, this hairpin points away from the parent CTD (*Wang et al., 2008*). *Cis* and *trans* DnaB monomers alternate around the hexameric ring. Indeed, this feature gives rise to the unusual six-fold/three-fold arrangement of sub-domains in the complete hexamer. Notably, in the DnaB configuration found in the BP complex, we find the following pattern of isomers in the protomers of DnaB from the bottom of the spiral to the top: chain A (*cis*), chain B (*trans*), chain C (*cis*), chain D (*trans*), chain E (*cis*), chain F (*trans*). However, in the ssDNA complex, we observe that the pattern of isomers differs: chain A (*trans*), chain B (*cis*), chain C (*trans*), chain D (*cis*), chain E (*trans*), chain F (*cis*). Thus, the mapping of CTDs between the two spiral forms of DnaB in closing Scheme I implies that a substantial rearrangement of the NTD tier must occur on helicase closing; this change consists of a *cis* to *trans* isomerization of DnaB subunits. In this scheme, closing is accompanied by disruption of each dimer interface in the NTD tier, a rotation of each isolated NTD, followed by re-establishment of the dimer, but with an adjacent, and distinct, DnaB subunit. However, the extensive interface found in the NTD tail to tail dimer (BSA =∼1800 Å$^2$ and (*Wang et al., 2008*; *Arias-Palomo et al., 2013*)) is expected to specify a stable interaction, and, thus, pose energetic challenges to closing Scheme I.

A second closing scheme can be envisioned through an alternate mapping of DnaB spirals as follows: BP: chain A → ssDNA complex: chain B, BP: chain B → ssDNA complex: chain C, etc (*Figure 8—figure supplement 2*). By this scheme, closing of the helicase involves dissociation of the CTD from the chain A of the BP complex, followed by translocation from the 'bottom' of the spiral to its new position at the 'top' of the spiral; presence of ADP in BP chain A could facilitate dissociation. Moreover, binding of ATP by the unfilled site (BP chain B) could stabilize the newly formed interaction with chain A after it has translocated. Also, owing to its presence at the breached interface, the CTD, and its associated linker helix, at the 'bottom' of the spiral makes the fewest contacts to neighboring subunits; these features could also enable dissociation and translocation to the 'top' of the spiral. Following translocation of the chain A CTD, the pattern of DnaB isomers in the BP complex matches that in the ssDNA complex. Several features of this scheme are attractive: first, the CTD motion required to translocate from the 'bottom' of the spiral to the 'top' is consistent with that in the 'hand-over-hand' mechanism for translocation in the 5' to 3' direction along ssDNA

proposed for DnaB (*Itsathitphaisarn et al., 2012*). Second, sealing of the breach in the NTD layer is achieved without the *cis*-to-*trans* isomerization of DnaB monomers, and concomitantly the energetically costly disruption of NTD dimers, required of scheme I. A similar scheme to closing scheme II has been proposed for closing the helicase in the *E. coli* DnaB-DnaC complex (*Arias-Palomo et al., 2013*). A more precise description of the mechanism of closing of the DnaB helicase awaits results of additional experiments.

As well as sealing the breaches in DnaB, the NTD layer undergoes an additional change: expansion of the central chamber from the constricted to the dilated state. Both the NTD and CTD layer in the BP complex populate the constricted state, however, in the ssDNA complex, DnaB is found in a hybrid state, with the CTD layer in the constricted state and the NTD layer in the dilated state. In this context, we note that the dilated, but not the constricted form, of DnaB is competent to interact with DNA primase (*Bailey et al., 2007a*; *Strycharska et al., 2013*). Transition from Stages II/III to Stage IV poises the DnaB helicase to interact with DNA primase (*Bailey et al., 2007a*; *Strycharska et al., 2013*), a critical step prior to assembly of the replisome. In this context, the constricted state of the NTD layer of DnaB in Stages II/III may serve to ensure that downstream events associated with replisome assembly, which begins with recruitment of DNA primase to a dilated NTD, do not take place until helicase loading at origin DNA has completed.

# Materials and methods

## Key resources table

| Reagent type (species) or resource | Designation | Source or reference | Identifiers | Additional information |
|---|---|---|---|---|
| Chemical compound, drug | Isopropyl-β-D-thiogalactoside (IPTG) | Sigma-Aldrich | 367-93-1 | |
| Chemical compound, drug | Tris(hydroxymethyl) aminomethane (Tris) | Sigma-Aldrich | 77-86-1 | |
| Chemical compound, drug | Glycerol | Fisher | 56-81-5 | |
| Chemical compound, drug | 1,4-Dithiothreitol (DTT) | Sigma-Aldrich | 3843-12-3 | |
| Chemical compound, drug | Sodium chloride | Fisher | 7647-14-5 | |
| Chemical compound, drug | Adenosine 5'-triphosphate (ATP) | Sigma-Aldrich | 34369-07-8 | |
| Chemical compound, drug | Magnesium chloride | Sigma-Aldrich | 7786-30-3 | |
| Chemical compound, drug | HEPES | Sigma-Aldrich | 75277-39-3 | |
| Chemical compound, drug | Potassium phosphate dibasic | Sigma-Aldrich | 7778-77-0 | |
| Chemical compound, drug | Potassium phosphate monobasic | Sigma-Aldrich | 7758-11-4 | |
| Chemical compound, drug | Potassium chloride | Fisher | 7747-40-7 | |
| Chemical compound, drug | 4-Morpholineetha nesulfonic acid (MES) | Sigma-Aldrich | 1266615-59-1 | |
| Chemical compound, drug | DSS (disuccinimidyl suberate) | Fisher | A39267 | |
| Chemical compound, drug | Ethylenedia minetetraacetic acid (EDTA) | Sigma-Aldrich | 60-00-4 | |
| Commercial assay or kit | HiTrap Q Fast Flow | GE Healthcare Life Sciences | 17-1153-01 | |

*Continued on next page*

*Continued*

| Reagent type (species) or resource | Designation | Source or reference | Identifiers | Additional information |
|---|---|---|---|---|
| Commercial assay or kit | HiTrap Heparin Fast Flow | GE Healthcare Life Sciences | 17-0406-01 | |
| Commercial assay or kit | Methyl Hydrophobic Interaction Chromatography | BioRad | 156–0080 | |
| Commercial assay or kit | Superdex 200 | GE | 17-1043-01 | |
| Commercial assay or kit | Zeba microspin desalting columns | Thermo Scientific | | |
| Strain, strain background (*Escherichia coli*) | BL21(DE3) | Novagen | 69450–3 | |
| Strain, strain background (*Escherichia coli*) | Rosetta | Novagen | 70954–3 | |
| Recombinant DNA reagent | pET24a-*Ecoli*-DnaB | Bacterial expression vector for the E. coli DnaB helicase | N/A | |
| Recombinant DNA reagent | pCDF-LP | Bacterial expression vector for the LP helicase loader from bacteriophage lambda | N/A | |
| Recombinant DNA reagent | pET24a-AA-DnaB | Bacterial expression vector for the Aquifex aeoliucs DnaB helicase | N/A | |
| Recombinant DNA reagent | pET24a-AA-DnaC | Bacterial expression vector for the Aquifex aeoliucs DnaC helicase loader | N/A | |
| Software, algorithm | Appion | (*Voss et al., 2010*) | | |
| Software, algorithm | Appion-Protomo | (*Noble and Stagg, 2015*) | http://appion.org | |
| Software, algorithm | CCP4 | (*Winn et al., 2011*) | http://www.ccp4.ac.uk | |
| Software, algorithm | COOT | (*Emsley et al., 2010*) | https://www2.mrc-lmb.cam.ac.uk/ personal/pemsley/coot/ | |
| Software, algorithm | CTFFind4 | (*Rohou and Grigorieff, 2015*) | http://grigorifflab.janelia.org/ctffind4 | |
| Software, algorithm | Cryosparc | (*Punjani et al., 2017*) | https://cryosparc.com | |
| Software, algorithm | Dali Server | (*Holm and Rosenström, 2010*) | http://ekhidna.biocenter.helsinki.fi/dali_server | |
| Software, algorithm | Dynamo | (*Castaño-Díez et al., 2012*) | https://wiki.dynamo.biozentrum.unibas.ch/w/index.php/Main_Page | |
| Software, algorithm | EMAN2 | (*Tang et al., 2007*) | http://blake.bcm.tmc.edu/EMAN2/ | |
| Software, algorithm | gAutomatch | | http://www.mrc-lmb.cam.ac.uk/kzhang/Gautomatch/ | |
| Software, algorithm | gCTF | (*Zhang, 2016*) | http://www.mrc-lmb.cam.ac.uk/kzhang/Gctf/ | |

*Continued on next page*

*Continued*

| Reagent type (species) or resource | Designation | Source or reference | Identifiers | Additional information |
|---|---|---|---|---|
| Software, algorithm | LEGINON | (*Suloway et al., 2005*) | http://emg.nysbc.org/redmine/projects/leginon/wiki/Leginon_Homepage | |
| Software, algorithm | LSQMan | (*Kleywegt, 2007*; *Kleywegt and Jones, 1997*) | http://xray.bmc.uu.se/usf/ | |
| Software, algorithm | MOLREP | (*Vagin and Teplyakov, 2010*) | http://www.ccp4.ac.uk/html/molrep.html | |
| Software, algorithm | MotionCor2 | (*Li et al., 2013*) | http://msg.ucsf.edu/em/software/motioncor2.html | |
| Software, algorithm | PDBeFold Server | (*Krissinel and Henrick, 2004*) | http://www.ebi.ac.uk/msd-srv/ssm/ | |
| Software, algorithm | PHENIX | (*Adams et al., 2010*) | http://www.phenix-online.org/ | |
| Software, algorithm | PyMOL | (*Schrodinger LLC, 2017*) | http://www.pymol.org | |
| Software, algorithm | Relion | (*Scheres, 2012*) | https://www2.mrc-lmb.cam.ac.uk/ relion/index.php/Main_Page | |
| Software, algorithm | ResMap | (*Kucukelbir et al., 2014*) | http://resmap.sourceforge.net/ | |
| Software, algorithm | SFCHECK | (*Vaguine et al., 1999*) | http://www.ccp4.ac.uk/html/sfcheck.html | |
| Software, algorithm | Swiss-Model | (*Biasini et al., 2014*) | https://swissmodel.expasy.org | |
| Software, algorithm | TOMO3D | (*Agulleiro and Fernandez, 2015*) | https://sites.google.com/site/3demimageprocessing/tomo3d | |
| Software, algorithm | UCSF Chimera | (*Pettersen et al., 2004*) | https://www.cgl.ucsf.edu/chimera/ | |

## Protein expression and purification

### *E. coli* DnaB helicase • phage λP loader

Bacterial expression of the isolated bacteriophage λP helicase loader under a variety of conditions yielded insoluble or poorly soluble material. As such, full length and truncated variants of the *E. coli* DnaB helicase • bacteriophage λP helicase loader complex were co-expressed in *E. coli* BL21(DE3) cells. Plasmids (*E. coli* DnaB: pET24-DnaB-EC, full length phage λP: pCDFDuet-λP, truncated λP containing residues 103–233: pCDFDuet-LPΔ102-NHis) were co-transformed into BL21(DE3). Standard methods were used to prepare starter cultures (*Sørensen and Mortensen, 2005*; *Terpe, 2006*; *Tolia and Joshua-Tor, 2006*), which were applied to a 10 L fermenter filled with superbroth media (24 gm $l^{-1}$ yeast extract, 12 gm $l^{-1}$ tryptone, 2.3 g $l^{-1}$ KH$_2$PO4, 12.5 g $l^{-1}$ K$_2$HPO4, 3.2% glycerol, 1 mM MgCl$_2$, and 0.1 mM CaCl$_2$) media supplemented with 20 µg ml$^{-1}$ kanamycin and 10 µg ml$^{-1}$ streptomycin. Fermentation was allowed proceed at 37° C to an OD$_{600}$ = 3 at a stir rate of 450 RPM and oxygen flow of 0.5 L min$^{-1}$. At the three-hour point, protein expression was induced by bringing the culture to 0.5 mM isopropyl-thio-galactipyranoside (IPTG); induction was allowed to proceed for 5 hr at 37°C. This procedure typically yielded 200–300 g of cells. Harvested cells were resuspended in 50 mM Tris-HCl pH 7.6, 10% (w/v) sucrose, 500 mM NaCl, 2 mM DTT, 10 mM MgCl$_2$ at a ratio of 5 ml per 1 g cells. The resuspended cells were flash frozen in liquid nitrogen and stored at −80°C until use.

Cells expressing full-length or truncated constructs of the *E. coli* DnaB helicase • bacteriophage λP complex were lysed using a French press. After lysis, cell debris was removed by centrifugation. The BP complex in the resulting soluble fraction was precipitated by addition of 0.2 mg ml$^{-1}$ ammonium sulfate and a 30 min incubation. The BP complex was resolved from both uncomplexed DnaB

and λP using a combination of cation (Q-sepharose, GE Healthcare), affinity (Heparin sepharose, GE Healthcare) and hydrophobic interaction (Methyl HIC, Bio-Rad) chromatography. The soluble isolated λP that emerged from the first step of chromatography was unstable and precipitated after coming off the column. All chromatography buffers used to prepare the BP complex contained 0.5 mM ATP. Purified BP complex (*Figure 2—figure supplement 1C*) was dialyzed into 20 mM Na-HEPES pH 7.5, 450 mM NaCl, 5% glycerol, 2 mM DTT, 0.5 mM MgCl$_2$, 0.5 mM ATP and concentrated to 18 mg ml$^{-1}$ by ultrafiltration. Purified BP complex was flash frozen in liquid nitrogen and stored at −80°C until use.

To prepare BP complexes with truncated λP (residues 103–233), pCDFDuet-λPΔ102-NHis and pET24-DnaB-EC were co-transformed into BL21 (DE3) cells and grown in LB media supplemented with 20 μg ml$^{-1}$ kanamycin and 10 μg ml$^{-1}$ streptomycin. Protein expression was induced with 0.5 mM IPTG, and induction was allowed to proceed for 4 hours at 37° C. Cells were harvested by centrifugation and lysed. DnaB•λPΔ102 in the soluble fraction was precipitated by treatment with 0.2 mg ml$^{-1}$ ammonium sulfate for 30 min. The resulting precipitate was collected by centrifugation and resuspended in 50 mM sodium phosphate pH 8.0, 450 mM NaCl, 5% glycerol, 0.5 mM MgCl$_2$, 0.1 mM ATP and 10 mM imidazole. The DnaB•λPΔ102 complex was purified by a combination of Ni-NTA affinity chromatography, cation (Q-Sepharose, GE Healthcare), and hydrophobic interaction (Methyl HIC, Bio-Rad) chromatography. Purified DnaB•λPΔ102 was dialyzed into 20 mM Na-HEPES pH 7.5, 450 mM NaCl, 5% glycerol, 2 mM DTT, 0.5 mM MgCl$_2$ and 0.1 mM ATP, concentrated with a Corning SpinX UF-20 100 kDa MWCO concentrator to ~4 mg ml$^{-1}$, flash frozen in liquid nitrogen, and stored at −80°C until use.

All protein chromatographic steps were performed using standard techniques (*Dunn et al., 2003*). Unless otherwise indicated all chromatographic steps were carried out at 4°C. The sequences of all genetic constructs used in this study were verified by DNA sequencing (not shown). Mass spectrometric analyses of bands from SDS-PAGE gels corresponding to the *E. coli* DnaB and the λP proteins confirmed the identity of both proteins (data not shown). All purification buffer components are listed in *Table 4*.

## *A. aeolicus* DnaB helicase • DnaC loader

To prepare complexes for analyses, *A. aeolicus* (AA) DnaB or *A. aeolicus* DnaC were expressed in *E. coli* Rosetta cells (Novagen) cultured at 37°C in Luria-Bertani (LB) broth supplemented with 50 μg ml$^{-1}$ kanamycin and 34 μg ml$^{-1}$ chloramphenicol.

**Table 4.** Protein purification buffers.

| Protein | Purification step | Buffer |
|---|---|---|
| BP | Cell Lysis | 50 mM Tris-HCl pH 7.6, 10% (w/v) sucrose, 500 mM NaCl, 2 mM DTT, 10 mM MgCl$_2$ |
| BP | Q-sepharose Chromatography | **Q-0**: 20 mM Tris pH 7.6, 5% glycerol, 1 mM DTT, 5 mM MgCl$_2$, 0.1 mM ATP<br>**Q-A**: 20 mM Tris pH 7.6, 5% glycerol, 1 mM DTT, 5 mM MgCl$_2$, 0.1 mM ATP, 50 mM NaCl<br>**Q-B**: 20 mM Tris pH 7.6, 5% glycerol, 1 mM DTT, 5 mM MgCl$_2$, 0.1 mM ATP, 1 M NaCl |
| BP | Heparin Chromatography | **H-0**: 10 mM HEPES pH 7.0, 5% glycerol, 1 mM DTT, 5 mM MgCl$_2$, 0.1 mM ATP<br>**H-A**: 10 mM HEPES pH 7.0, 5% glycerol, 1 mM DTT, 5 mM MgCl$_2$, 0.1 mM ATP, 50 mM NaCl<br>**H-B**: 10 mM HEPES pH 7.0, 5% glycerol, 1 mM DTT, 5 mM MgCl$_2$, 0.1 mM ATP, 1 M NaCl |
| BP | Methyl Chromatography | **M-A**: 20 mM Tris-HCl pH 8.2, 5% glycerol, 1 mM DTT, 5 mM MgCl$_2$, 0.1 mM ATP, 1 M (NH$_4$)$_2$SO$_4$<br>**M-B**: 20 mM Tris-HCl pH 8.2, 5% glycerol, 1 mM DTT, 5 mM MgCl$_2$, 0.1 mM ATP |
| BP | Dialysis | 20 mM Na-HEPES pH 7.5, 450 mM NaCl, 5% glycerol, 2 mM DTT, 0.5 mM MgCl$_2$, 0.5 mM ATP |
| BC | Cell Lysis | 50 mM potassium phosphate pH 7.6, 500 mM KCl, 10% glycerol, 10 mM β-mercaptoethanol |
| BC | Heparin Sepharose Chromatography | **H-0**: 20 mM MES pH 6.0, 5% glycerol, 1 mM DTT<br>**H-A**: 20 mM MES pH 6.0, 50 mM KCl, 5% glycerol, 1 mM DTT<br>**H-B**: 20 mM MES pH 6.0, 1 M KCl, 5% glycerol, 1 mM DTT |
| BC | Q-sepharose Chromatography | **Q-0**: 20 mM Tris pH 8.7, 10% glycerol, 10 mM MgCl$_2$<br>**Q-A**: 20 mM Tris pH 8.7, 10 mM KCl, 10% glycerol 10 mM MgCl$_2$<br>**Q-B**: 20 mM Tris pH 8.7, 1 M KCl, 10% glycerol, 10 mM MgCl$_2$ |
| BC | Size Exclusion Chromatography | 20 mM MES pH 6.0, 500 mM KCl, 10% glycerol, 1 mM DTT, 0.1 mM EDTA |

DOI: https://doi.org/10.7554/eLife.41140.027

Cells that expressed AA DnaB were harvested and resuspended in 50 mM potassium phosphate pH 7.6, 500 mM KCl, 10% glycerol, 10 mM β-mercaptoethanol at a ratio of 5 ml per 1 g cells. Cells that expressed AA DnaC were harvested and resuspended in 50 mM HEPES pH 7.5, 50 mM potassium glutamate, 10 mM magnesium acetate, 5 mM β-mercaptoethanol at a ratio of 5 ml per 1 g cells. The resuspended cells were flash frozen in liquid nitrogen and stored at −80°C until use.

Cells in which AA DnaB and AA DnaC had been separately expressed were co-lysed at a ratio of 1 g DnaB: 2 g DnaC. The co-lysis approach was adopted because, although over-produced in *E. coli* in soluble form, instability of AA DnaC precluded efforts to work with the isolated protein, however, the AA BC complex could be readily purified. Inclusion of an excess of DnaC maximized the amount of the AA DnaB•DnaC (BC) complex in our preparation. Cells were lysed by sonication (total time: 4.5 min, with pulses of 0.66 s on/0.33 s off at 60% amplitude with Sonic Dismembrator sonicator (Fisher Scientific). The soluble fraction from this procedure was incubated at 65°C for 30 min; the AA BC complex remained in solution after removal of the precipitated material by centrifugation. The AA BC complex was further purified by a combination of affinity (Heparin sepharose Fast Flow, GE Healthcare), cation exchange (Q-sepharose Fast Flow, GE Healthcare), and size exclusion (Superdex-200, GE Healthcare). Purified AA BC complex was exchanged into 20 mM MES pH 6.0, 500 mM KCl, 10% glycerol, 1 mM dithiothreitol (DTT), 0.1 mM ethylenediaminetetraacetic acid (EDTA), flash frozen in liquid nitrogen, and stored at −80°C until use. Unless otherwise indicated all chromatographic steps were carried out at 4°C. All purification buffer components are listed in *Table 4*.

## Grid preparation for Cryo-EM and Cryo-ET

400 mesh carbon grids with holey carbon (0.6/1.0, Quantifoil, Großlöbichau, Germany) were coated with ~50 nm of gold by evaporating 30 cm of 0.2 mm Au wire (EMS) onto 50 grids using an Edwards Auto306 evaporator and the carbon layer subsequently removed by plasma cleaning for 5 min in a Gatan Solarus plasma cleaner (Gatan, Pleasanton, CA). Prior to sample adhesion, grids were plasma cleaned at 70% power, with a gas flow of 30% (75 parts argon, 25 parts oxygen) for 60 s using a NanoClean model 1070 (Fischione Instruments).

BP complex was freshly thawed on ice and diluted to a concentration of 1.5 μM in 20 mM Na-HEPES pH 7.5, 450 mM NaCl, 2 mM DTT, 0.5 mM MgCl$_2$, 0.5 mM ATP. 3.0 μL of the resulting solution was pipetted onto a fresh plasma cleaned grid; the sample was allowed to adsorb for 30 s at 100% humidity and 4°C, blotted for 3 s with a blot force of 4 and plunge frozen in liquid nitrogen-cooled liquid ethane. Sample adsorption and blotting were performed using a Vitrobot Mark IV (FEI, Hillsboro, Oregon). All grids were stored in liquid nitrogen until data acquisition.

## Single-particle cryo-EM and cryo-ET image acquisition

Grids were loaded into a Titan Krios (FEI, Hillsboro, Oregon), fitted with a Gatan K2 Summit (Gatan, Pleasanton, California) direct electron detector, operating at an acceleration voltage of 300 kV. Single particle movies were recorded at a pixel size of 1.07 Å with automatic hole targeting using LEGI-NON software suite (*Suloway et al., 2005*). Images were recorded for 10 s at a frame rate of 0.2 s in counting mode with a dose rate of 8.0 e⁻/Å$^2$ s$^{-1}$, an accumulated sample dose of 65 e⁻/Å$^2$, and a defocus range of −1.0 to −3.0 μm. Three 24 hr sessions produced 2426 micrograph movies. (Additional details may be found in the Appendix section).

In addition, five tilt series were collected from the same grids as used for the single particle collection. To minimize sample variation, tomography data were collected during a session that immediately following the single particle data collection described above. Tilted images were collected bi-directionally over a tilt range of −45° to +45° in 3° increments with a dose of 2.57 to 3.3 e⁻/Å$^2$ per tilt increment (subdivided over seven to nine frames) and a total accumulated sample dose of 90 e⁻/Å$^2$. Data were collected with a pixel size of 1.76 Å and at defocus values of −2.8 μm, −6.1 μm, and −9.3 μm.

## CryoET image analysis and reconstruction

The tilt series of images was aligned using a fiducial-less algorithm as implemented in Appion-Protomo (*Winkler and Taylor, 2006*; *Noble and Stagg, 2015*). Tomo3D (*Agulleiro and Fernandez, 2011*; *Agulleiro and Fernandez, 2015*) was used to reconstruct tomograms from the aligned images. ~ 1000 particles were picked from the resulting tomograms (4 × 4 binned) and aligned in

Dynamo (*Castaño-Díez et al., 2012*; *Castaño-Díez, 2017*; *Castaño-Díez et al., 2017*) to produce a sub-tomogram average of the BP complex. This sub-tomogram average was then used as an initial model for single-particle analysis (*Castaño-Díez et al., 2012*; *Castaño-Díez, 2017*; *Castaño-Díez et al., 2017*) and as a template for automatic particle picking. EMAN2 (*Tang et al., 2007*; *Bell et al., 2016*; *Ludtke, 2016*) was used to generate 2D projections of the BP complex sub-tomogram average in 30 evenly spaced viewing directions. These 2D projections were then used as templates for template-based picking of the micrographs using Gautomatch (version 0.50, http://www.mrc-lmb.cam.ac.uk/kzhang/Gautomatch/). Additional details of our cryoET study are available here: (*Noble et al., 2018*).

## CryoEM image analysis and reconstruction

The frames of the 2426 micrograph movies were frame aligned for whole-frame motion correction with MotionCorr (*Li et al., 2013*) in Appion (*Voss et al., 2010*). Following frame alignment, the contrast transfer functions were estimated by CTFFind4 (*Rohou and Grigorieff, 2015*) and gCTF (*Zhang, 2016*). In order to retain the highest quality set of particles, gCTF-corrected micrographs that showed a CTF estimate of $\leq$10 Å at a confidence cutoff of 0.8 (*Zhang, 2016*) were included in the procedure described below. ~267,000 particles were picked with Gautomatch, using 2D templates derived from the cryo-ET model (see above and Appendix). Accuracy of particle picking procedures was assessed via two-dimensional classification by random sampling of a subset of 26,500 particles in Relion (*Scheres, 2012*; *Kimanius et al., 2016*; *Fernandez-Leiro and Scheres, 2017*) (*Figure 2—figure supplement 2*). The entire particle stack was further classified in three dimensions into eight classes using Relion with C1 symmetry for assessing particle quality (*Figure 2—figure supplement 4*). The majority of the particles were found in 4 of 8 classes. The remaining 25.7% of the particles were found in 3D classes and did not appear to contain intact particles. Alignment parameters of class 2 (33.2% of particles), class 3 (11.5% of particles), class 6 (16.9% of particles) and class 7 (12.6% of particles) were further assessed by 2D classification, and subsequently refined in Cryosparc (*Punjani et al., 2017*). Notably, only the particles in class two produced a final volume with a resolution beyond 8 Å; all other classes resulted in deformed volumes. Class 2, which contained 91,728 particles, was then subjected to another round of 2D classification resulting in a cleaner set containing 90,883 particles, which were subsequently refined in Relion and produced a map with a global resolution of 4.1 Å. Map resolution was assessed by Fourier shell correlation (FSC) of independently refined half sets using the gold standard 0.143 value as the cut-off criteria (*Henderson et al., 2012*; *Rosenthal and Rubinstein, 2015*). The final map was generated after post-processing in Relion with a mask set at a threshold of 0.008; the threshold was determined by inspection of the EM volume in UCSF-CHIMERA (*Pettersen et al., 2004*). Our goal in determining the threshold for the mask was to ensure that domains of the complex were included and that noise was omitted. Local resolution was estimated using RESMAP (*Kucukelbir et al., 2014*), and showed a higher overall resolution for parts of the map that correspond to DnaB (~80% of the volume) and slightly lower resolution for portions of the map (~20%) that correspond to λP (*Figure 2—figure supplement 3C,D*). In the Appendix, we detail additional efforts to identify whether BP conformers others than the one described above were found in our data set.

## Model building

UCSF-CHIMERA (*Pettersen et al., 2004*) and SFCHECK (*Vaguine et al., 1999*) in the CCP4 program suite (*Winn et al., 2011*) were used to establish the chirality of the EM-derived density map (EMDB: EMD-7076). Since no atomic model of *E. coli* DnaB was available, we used the Swiss-Model web-hosted software (*Biasini et al., 2014*), https://swissmodel.expasy.org) to calculate a homology model. To facilitate eventual refinement of our model against EM density maps, the homology model was constructed out of the highest resolution structures available for the DnaB NTD (PDB: 2R5U, (*Biswas and Tsodikov, 2008*)) and CTD (PDB: 3BH0, (*Wang et al., 2008*)). Six instances of the homology models for the NTD and the CTD sub-structures of *E. coli* DnaB were unambiguously placed into our map by MOLREP (*Vagin and Teplyakov, 2000*; *Vagin and Teplyakov, 2010*). In addition, the map enabled us to model five of six NTD-CTD linker segments (*Figure 2—figure supplement 6C,D*). As a result, we could unambiguously connect each NTD to its cognate CTD in the complete model of DnaB in the BP complex. Understanding relationships between each NTD and

CTD enabled informative comparisons to be made with the closed planar (PDB = 4NMN) and closed spiral forms (PDB = 4ESV) of DnaB. Clear density was observed for five ADP molecules at the expected sites on DnaB; the nucleotide site formed by the CTDs that line the ruptured interface is vacant (*Figure 2—figure supplement 6B*). Protein chains in the DnaB portion of the model were named following the example of the ssDNA complex (*Itsathitphaisarn et al., 2012*) in which Chain A is at the 'bottom' of the spiral, and chain B is at the 'top' of the spiral. As noted main text, 'top' and 'bottom' of the DnaB spiral are separately defined in the BP and ssDNA complexes by consideration of the position of the CTDs alone. Also, the terms 'top' and 'bottom', as we use them here, are inverted from that in reference (*Itsathitphaisarn et al., 2012*).

No atomic level information for the λP loader has been previously described. As such, an atomic model (~122/233 Cα) for the five λP protomers was constructed manually. The current resolution of our EM map did not allow the chain direction of λP to be determined, nor could amino acid side chains to be assigned to the structure. To address this limitation, we used CX-MS and protein-binding studies to determine that our EM maps included the carboxy-terminal domain of λP (Appendix and *Figure 2 – Figure 2—figure supplement 1*, *Figure 3—figure supplement 1*). We have named the five λP protomers as P1 (chain Z), P2 (chain Y), P3 (chain X), P4 (chain W) and P5 (chain V) (*Figure 2C,D*). As described in the Appendix, we have numbered λP based on the assumption that the last Cα observed corresponds to residue 233. However, the implied assignment of sequence to structure should be considered tentative.

Inspection of our EM maps at lower contour (four sigma in PyMOL (*Schrodinger LLC, 2017*)) revealed additional weak EM density adjacent to the λP1 protomer (chain Z). Into this density, we could build an additional ~75 amino acids (*Figure 2—figure supplement 3E,F*). Included in this segment is a rod-like density, which contacts the CTD of DnaB (chain B) and lines the breached interface; this segment could not be accounted by the known structure of DnaB (Appendix). We speculate that this segment represents the amino-terminal domain of a λP protomer (likely λP1, chain Z). Presence of a segment of λP at the breached interface signals that additional mechanisms of helicase opening remain to be discovered.

Model building and visualization was performed in COOT (*Emsley et al., 2010*).

## Model refinement

The BP model was refined using the real_space_refine routine in PHENIX (*Afonine et al., 2018b*). The refinement converged at a correlation coefficient of 0.739. The final model (PDB: 6BBM) contains residues (chain A: 27–166 and 204–468, chain B: 18–464, chain C: 19–471, chain D: 17–471, chain E: 18–471, chain F: 21–468, chain V: 109–233, chain W: 109–233, chain X: 113–233, chain Y: 110–233 and chain Z: 111–233) with five ADP molecules. Notably, the extended λP1 segment described above was omitted from the final model owing to its weak occupancy. Analysis of the final model using the Ramachandran plot, as implemented in Phenix, revealed that our model exhibited 84.5% of the residues in the favored region, 15.3% in the allowed region, and 0.2% in the in the outlier region. Data collection and refinement statistics can be found in *Table 1*. A map to model FSC was generated using the Mtriage option of the Phenix suite and agreed with global resolution estimates (*Figure 2—figure supplement 3B*).

## Model analysis

Structural analysis and visualization were carried out using the CCP4 software package (*Winn et al., 2011*), the Uppsala software suite (*Kleywegt et al., 2001*; *Sierk and Kleywegt, 2004*; *Kleywegt, 2007*), UCSF-CHIMERA (*Pettersen et al., 2004*), Phenix (*Adams et al., 2010*; *Afonine et al., 2018a*), and PyMOL (*Schrodinger LLC, 2017*).

Analysis of the angular relationship between the NTD and CTD tiers of the various forms of DnaB was performed by fitting a plane to six points corresponding to the centers of gravity of the six CTD domains (Moleman2, Uppsala software suite (*Kleywegt and Jones, 1997*; *Kleywegt, 2007*)). The plane fitting procedure was repeated for the centers gravity of the six NTD globe domains (residues: *A. aeolicus*: 14–96; *E. coli*: 31–113; *G.st*: 14–96). The angle between the resulting planes from the NTD and CTD layers of DnaB was calculated as the arc-cosine of the quotient of the dot product and the cross product of the normal vectors of these planes.

## Native mass spectrometry

Native mass spectrometry (MS) of all samples (apo-BP, ssDNA-BP, *A. aeolicus* DnaB-DnaC) was carried out at a protein concentration of 5 μM. ssDNA (Genewiz, LLC) in our experiments were of two types: (1) a 43-mer sequence (5' TGACGAATAATCTTTTCTTTTTTCTTTTGTAATAGTGTCTTTT 3') derived from DNA unwinding element (DUE) of Oriλ (*Learn et al., 1997*) and (2) a series of thymidylate homopolymers of varying nucleotide length (T 25 nt, T 35 nt or T 45 nt). Protein samples were buffer-exchanged into native MS solution using Zeba microspin desalting columns (Thermo Scientific) with a 40 kDa molecular weight cut-off (MWCO). The MS buffer contained 450 mM ammonium acetate, pH 7.5, 0.5 mM magnesium acetate, 0.01% Tween-20, and was selected to mimic the ionic strength of that in the cryo-EM samples. For samples containing ssDNA, the buffer-exchanged BP complex was incubated with nucleic acid at a molar ratio of 1:1.2 for 30 min on ice prior to native MS experiments. A 2–3 μL aliquot of the buffer-exchanged sample was loaded into an in house fabricated gold-coated quartz capillary. The sample was then sprayed into an Exactive Plus EMR instrument (Thermo Fisher Scientific) using a static nanospray source. The MS parameters used are spray voltage, 1.0–1.4 kV; capillary temperature, 100°C; in-source dissociation, 10 V; S-lens RF level, 200; resolving power, 8750 at $m/z$ of 200; AGC target, $0.5–3 \times 10^6$; number of microscans, 5; maximum injection time, 200 ms; injection flatapole, 8 V; interflatapole, 4 V; bent flatapole, 4 V; high energy collision dissociation (HCD), 200 V; ultrahigh vacuum pressure, $8–9 \times 10^{-10}$ mbar; total number of scans, 100. The EMR instrument was mass calibrated using cesium iodide. The native MS spectra were visualized using the Thermo Xcalibur Qual Browser (version 3.0.63). Deconvolution was performed manually. For samples buffer-exchanged into native MS solution containing magnesium acetate, the deviations from expected mass ranged from 0.04% to 0.09% due to peak broadening from nonspecific magnesium adduction (see *Table 3* for comparisons with mass measurements taken with protein samples without magnesium acetate). All mass spectrometric data were measured in the laboratory of Professor Brian Chait of The Rockefeller University.

## Crosslinking mass spectrometry

*Ec*DnaB-λP-Oriλ derived ssDNA (0.2 mg ml$^{-1}$, with a protein to DNA ratio of 1:1.2) was cross-linked with 2 mM disuccinimidyl suberate (DSS) overnight at 4°C; the crosslinking reaction was quenched with 50 mM ammonium bicarbonate. Isolated λP was not analyzed since it could not be produced in soluble form. We observed that crosslinking was more efficient when ssDNA was included in the crosslinking reaction. This factor, in combination with the finding that preparations of the BP•ssDNA complex contained a single entity (*Figure 3*), led us to analyze the ssDNA containing crosslinked sample; we reasoned that resulting MS data would arise from a more homogeneous preparation. Cross-linked samples were reduced with 25 mM DTT for 10 min at 70°C, alkylated with 100 mM 2-chloroacetamide at room temperature in the dark for 30 min, then separated by SDS-PAGE with a 3–8% Tris-Acetate gel (NuPAGE, Thermo Fisher Scientific), and stained with Coomassie-blue for visualization. The region containing proteins between 170 kDa and 460 kDa was excised, crushed, treated with trypsin overnight to generate cross-linked peptides as previously described (*Shi et al., 2014*; *Shi et al., 2015*). Peptides were desalted and concentrated on C18 solid phase extraction material (Empore), loaded onto an EASY-Spray column (Thermo Fisher Scientific ES800: 15 cm ×75 μm ID, PepMap C18, 3 μm) via an EASY-nLC 1200 and gradient-eluted for online ESI–MS and MS/MS analyses with a Q Exactive Plus mass spectrometer (Thermo Fisher Scientific). MS/MS analyses of the top eight precursors in each full scan used the following parameters: resolution: 17,500 (at 200 Th); AGC target: $2 \times 10^5$; maximum injection time: 800 ms; isolation width: 1.4 m/z; normalized collision energy: 24%; charge: 3–7; intensity threshold: $2.5 \times 10^3$; peptide match: off; dynamic exclusion tolerance: 1500 mmu. Cross-linked peptides were identified from mass spectra by pLink, and peptide-spectrum matches were manually verified as previously established (*Shi et al., 2014*; *Shi et al., 2015*). The crosslinked lysine pairs provided by the above procedure were used to evaluate the BP model as described in the Appendix.

## Acknowledgements

We are grateful to Roger McMacken for sharing unpublished data, to Mike O'Donnell for sharing expression plasmids, to the staff of the New York Structural Biology Center, including Bridget

Carragher, Clint Potter and to the staff of the CUNY Advanced Science and Research Center (ASRC), including Tong Wang. We thank Mike O'Donnell, Nicole Francis, and R. Blake Hill for critical reading of the manuscript. This work is dedicated to the memory of Professor Megan Davey.

Science in the Jeruzalmi lab was supported by the National Institutes of Health (R01 GM084162), the National Science Foundation (MCB 1818255), and the National Institute on Minority Health and Health Disparities (5G12MD007603-30). The Chait Lab is supported by the National Institutes of Health (P41 GM103314 and P41 GM109824). JC was the recipient of a fellowship from the U.S. Department of Education Graduate Assistance in Areas of National Need (GAANN) Program in Biochemistry, Biophysics, and Biodesign at The City College of New York (PA200A150068). AJN was supported by a grant from the NIH National Institute of General Medical Sciences (F32GM128303). Some of this work was performed at the Simons Electron Microscopy Center and National Resource for Automated Molecular Microscopy located at the New York Structural Biology Center, supported by grants from the Simons Foundation (SF349247), NYSTAR, and the NIH National Institute of General Medical Sciences (GM103310) with additional support from the Agouron Institute (F00316) and the NIH (OD019994). We thank Silicon Mechanics and their Research Cluster Grant program for award of a high-performance computing cluster that was used in this research (SM-2015–289297).

## Additional information

### Funding

| Funder | Grant reference number | Author |
| --- | --- | --- |
| National Institutes of Health | GM084162 | David Jeruzalmi |
| National Institutes of Health | 5G12MD007603-30 | David Jeruzalmi |
| Simons Foundation | SF349247 | Edward T Eng |
| Agouron Institute | F00316 | Edward T Eng |
| National Institutes of Health | OD019994 | Edward T Eng |
| Department of Education and Training | PA200A150068 | Jillian Chase |
| National Institutes of Health | P41 GM103314 | Brian Chait |
| National Institutes of Health | P41 GM109824 | Brian Chait |
| National Institutes of Health | F32GM128303 | Alex J Noble |
| National Science Foundation | MCB 1818255 | David Jeruzalmi |

The funders had no role in study design, data collection and interpretation, or the decision to submit the work for publication.

### Author contributions

Jillian Chase, Conceptualization, Data curation, Formal analysis, Validation, Investigation, Visualization, Writing—original draft, Writing—review and editing; Andrew Catalano, Martin Samuels, Investigation, Methodology; Alex J Noble, Data curation, Software, Formal analysis, Investigation, Methodology; Edward T Eng, Resources, Data curation, Formal analysis, Validation, Methodology; Paul DB Olinares, Resources, Data curation, Software, Formal analysis, Validation, Investigation, Methodology; Kelly Molloy, Conceptualization, Formal analysis, Validation, Investigation, Methodology; Danaya Pakotiprapha, Conceptualization, Formal analysis, Investigation; Brian Chait, Conceptualization, Data curation, Software, Formal analysis, Validation, Investigation, Methodology; Amedee des Georges, Formal analysis, Validation, Investigation, Methodology; David Jeruzalmi, Conceptualization, Formal analysis, Supervision, Funding acquisition, Validation, Investigation, Visualization, Methodology, Writing—original draft, Project administration, Writing—review and editing

### Author ORCIDs

Jillian Chase (iD) http://orcid.org/0000-0002-1027-6516
Alex J Noble (iD) http://orcid.org/0000-0001-8634-2279

Edward T Eng ⓘ http://orcid.org/0000-0002-8014-7269
Danaya Pakotiprapha ⓘ http://orcid.org/0000-0002-5017-8283
David Jeruzalmi ⓘ http://orcid.org/0000-0001-5886-1370

**Decision letter and Author response**
Decision letter https://doi.org/10.7554/eLife.41140.036
Author response https://doi.org/10.7554/eLife.41140.037

## Additional files

**Supplementary files**
• Transparent reporting form
DOI: https://doi.org/10.7554/eLife.41140.028

**Data availability**

Cryogenic electron microscopy maps have been deposited with the EMDB under accession number EMD-7076. Atomic coordinates have been deposited with the PDB under the accession code 6BBM.

The following datasets were generated:

| Author(s) | Year | Dataset title | Dataset URL | Database and Identifier |
|---|---|---|---|---|
| Chase J | 2018 | Mechanisms of Opening and Closing of the Bacterial Replicative Helicase | http://www.rcsb.org/structure/6BBM | Protein databank, 6BBM |
| Chase J, Catalano A, Noble AJ, Eng ET | 2018 | Mechanisms of Opening and Closing of the Bacterial Replicative Helicase | http://www.ebi.ac.uk/pdbe/entry/emdb/EMD-7076 | Electron Microscopy Data Bank, EMD-7076 |

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

## Appendix 1

DOI: https://doi.org/10.7554/eLife.41140.029

# Single-particle CryoEM and CryoET analysis of BP complex

Efforts to crystallize the full-length *Escherichia coli* DnaB • λP helicase loader complex produced multiple crystal forms. However, poor internal order made these specimens unsuitable for analysis using X-ray crystallography. We therefore turned to single particle analysis using cryogenic electron microscopy (cryo-EM). Analysis of cryo-EM images of the full-length BP complex (*Figure 2—figure supplement 1*) using Gaussian-based methods to identify particles resulted in high quality two-dimensional (2D) class averages that displayed excellent ratios of signal to noise (*Figure 2—figure supplement 2*). However, efforts to obtain a high resolution map from initial models calculated with both common-line approaches and existing structures (DnaB, PDB entries = 2R6D (*Bailey et al., 2007a*) and 4ESV (*Itsathitphaisarn et al., 2012*), DnaB•DnaC, EMD entry = 2322 (*Arias-Palomo et al., 2013*)) were unsuccessful. We thus turned to cryo-electron tomography (cryo-ET) to obtain a reliable and unbiased initial model, as the advent of motion correction and gold grids (*Russo and Passmore, 2014*) made sub-tomogram averaging of small complexes possible. (A detailed description of the cryoET methods that we used appears here: (*Noble et al., 2018*)). Analysis of five tomograms (~1000 particles) resulted in three low resolution ~40 Å initial models each of which resembled an oblong, cracked particle. Such an outcome would not have been suspected from the apparently symmetrical 2D averages obtained initially (*Figure 2—figure supplement 2*).

In an attempt to improve the precision of particle picking and potentially detect lower contrast poses of the complex, we used the cryoET initial model for template-based particle picking. 2D projections of the initial model using EMAN2 (*Tang et al., 2007*; *Bell et al., 2016*; *Ludtke, 2016*) enabled template-based identification of a larger set of 267,000 particles. With the subsequent 2D classification of a random subset of this dataset, it became immediately apparent that many new poses of the complex, which had not been previously identified within the cryo-EM dataset using Gaussian-based picking methods, most likely due to their lower contrast, were now present (*Figure 2—figure supplement 2*).

Template-based particle selection using the cryoET model 2D projections was performed with Gautomatch. Three-dimensional (3D) classification of this larger set of particles (n = 267,000, K = 8) using the tomography-derived initial model served to identify intact BP particles. Of these classes, four contained deformed particles and were discarded, the remaining four were pushed forward to auto-refinement. Only one of the four auto-refined classes, which contained 91,632 particles, produced a high-resolution map. This particle stack was subjected to 2D classification in order to remove any particles that contained DnaB alone; the result was a clean stack of 90,833 BP complex particles. Post-processing produced a map with an overall resolution of 4.1 Å (*Figure 2A and B*; *Figure 2—figure supplement 3* ). Local resolution estimates indicate higher local resolution for map regions attributed to DnaB (accounting for roughly ~80% of map), whereas regions interpreted as λP exhibited comparatively lower local resolution (*Figure 2—figure supplement 3C,D*).

Analysis of our 4.1 Å EM density map was aided by prior studies, which established the two-domain architecture of a DnaB protomer and described how its amino- (NTD) and carboxy-terminal domains (CTD) assemble into a two layer structure in the complete hexamer (*Bailey et al., 2007a*; *Bailey et al., 2007b*; *Wang et al., 2008*; *Lo et al., 2009*; *Strycharska et al., 2013*). As no atomic model of the *E. coli* DnaB ortholog was available, we built separate homology models for the NTD (residues 1–173) and CTD (residues 203–441) segments of DnaB using the highest resolution coordinates available (NTD: PDB 2R5U (*Biswas and Tsodikov, 2008*), CTD: PDB 3BH0 (*Wang et al., 2008*)). Six copies each of the resulting models of the NTD and CTD were unambiguously placed in our EM density map. The six protomers were named A, B, C, D, E, and F in keeping with the nomenclature of PDB

entry = 4ESV (*Itsathitphaisarn et al., 2012*). Although individual NTD and CTD domains fit well into our map, their arrangement in the BP complex differs significantly from previously described DnaB structures. We could also build five of the six NTD-CTD linkers (residues 174–203, chains B, C, D, E, F). This feature of our model established an unambiguous connection between each NTD and its cognate CTD in the model.

Our EM density map also showed clear density for 5 copies of the λP protein. In our maps, the structure of λP comprises a compact helical domain and an extended region terminated by an alpha helix. Since λP has no sequence homologs, nor is an atomic model available, this part of the density was built by hand. Notably, our model encompasses $\geq$50% (122/233 for λP1, 123/233 for λP2, 120/233 for λP3 and 124/233 for λP4 and λP5) of the expected number of alpha carbons for λP. We have termed the five λP protomers: λP1 (chain Z), λP2 (chain Y), λP3 (chain X), λP4 (chain W) and λP5 (chain V) (*Figure 2D*). At present, the resolution of our map in the λP region does not permit unambiguous mapping of the amino acid sequence to the structure, nor does the map allow for chain polarity to be established. As we discuss below, we used two orthogonal approaches: (1) binding studies with truncations of λP (*Figure 2—figure supplement 1*), and (2) cross-linking mass spectrometry (CX-MS, *Figure 3C and D* and *Figure 3—figure supplement 1*) to tentatively conclude that the segment of λP visible in our maps represents the carboxy-terminal segment.

Although the model for the BP complex revealed by EM was consistent with the predominant form ($B_6P_5$) observed in our native MS analysis (*Figure 3C*), we wondered whether other conformers or stoichiometric states were present. We, therefore, re-analyzed our EM data sets to address these questions. Template-based particle picking, as previously described, yielded 267,000 particles. 3D classification (K = 8) in Relion produced only one class that refined to high resolution (Class 2 with 91,728 particles). We therefore set out to determine: (1) Is class 2 is a mixture of BP entities with stoichiometries other than $B_6P_5$ (as implied by our native MS experiments)?; or (2) Can alternate sorting procedures reveal presence of other conformers of the $B_6P_5$ species in our sample?

To this end, we subjected the 91,728 particles in class 2 (the stack generated before 2D classification was used to remove any non-BP particles) to heterogeneous refinement, homogenous refinement, and 2D classification in Cryosparc. Refinement encompassed a K = 6 so that the six possible stoichiometric states of the BP complex, including $B_6P_1$, $B_6P_2$, $B_6P_3$, $B_6P_4$, $B_6P_5$, and $B_6P_6$ could potentially be isolated. Heterogenous refinement refers to a procedure in which six individual initial models were used to initialize refinement of each of the six classes into which particles were sorted. By contrast, homogenous refinement refers to a procedure in which a single initial model was used to initialize refinement with six classes. This procedure produced distorted 3D volumes with the exception of classes that corresponded to the $B_6P_5$ species (*Figure 2—figure supplement 5*). Particles that were not distributed into the main class produced volumes lacking density for λP entirely; interestingly, these volumes resembled deformed non-planar DnaB particles. Furthermore, 2D classification in Cryosparc of class 2 confirmed that the particles in this class were of high quality. We did note presence of a small number ($\leq$1.4%) of particles that appeared to correspond to the closed planar ring form of DnaB (*Figure 2—figure supplement 5*). However, elimination of these particles did not produce a higher resolution map, nor a map that was substantially different than the original.

We next interrogated our entire particle set (267,000 particles) for other conformers or stoichiometric states of the BP complex via 3D classification (K = 6, using our 4.1 Å BP map filtered to 40 Å as a single initial model) and subsequent refinement of each resulting class in Cryosparc. This analysis produced two additional density maps which showed density for λP, however, no additional conformers or stoichiometric states of the BP complex were observed (*Figure 2—figure supplement 5*). We note a slightly different positioning for density corresponding to the NTD of DnaB chain A between these two additional maps; this may indicate that this region of the molecule is flexible.

The above analyses led us to conclude that the $B_6P_5$ species reported by our EM density maps represents the only conformer and stoichiometric state in our sample. This view is supported by native MS experiments of both isolated BP and the BP•ssDNA complex

(*Figure 3C and D*). The finding that two orthogonal techniques converged on the same value points to a physiological stoichiometry of $B_6P_5$. We also conclude that the $B_6P_4$ and $B_6P_6$ entities observed as minor components by the native MS are either (1) artifacts of the co-expression procedure used to the prepare the BP complex, or (2) unstable intermediates that do not survive our grid preparation procedure.

## Establishment of the chain direction of λP in the BP complex

Efforts to build a complete atomic model of the λP helicase loader in the BP complex were stymied by the low resolution of portions of our density maps (*Figure 2—figure supplement 3E,F*), and the lack of relatives for λP in the structural database. Nevertheless, we were able to reliably build ~122 alpha carbons, which represents ~50% the complete sequence, and, less reliably, an additional 75 alpha carbons of one protomer (λP1, chain Z). However, neither the amino acid sequence nor the amino to carboxy (N to C) chain direction could be assigned with confidence from the density map alone.

To establish polarity of the λP chains in our model, we employed two orthogonal approaches: 1) a binding study with histidine tagged truncations of λP and 2) cross-linking mass spectrometry (CX-MS). Binding studies indicate that a carboxy-terminal construction of λP (residues 103–233) retains a robust interaction with DnaB (*Figure 2—figure supplement 1*). Pull-down experiments with the N-terminus of λP (residues 1–110) were stymied by the fact that this construct alone or when co-transformed with DnaB did not express under conditions attempted. The finding that the carboxy terminal segment of λP retained an interaction with DnaB, in combination with the observation of unresolved density adjacent to λP protomer λP1 (chain Z), which we attribute above to the amino-terminal domain of λP, led us to make the conservative assumption that the carboxy-terminal segment of λP is represented in the high contours of our EM map. As such, we fixed the direction of the protein chains corresponding to λP in the BP complex such that the last segment of visible density represents the carboxy-terminus; this segment is adjacent to the NTD-CTD linker helix. We have also tentatively assumed that the portion of λP built into the last segment of visible density represents the extremity of the amino acid sequence. Even though our model for λP contains no side chains, the assumptions made above imply an assignment of sequence to structure; this assignment should be considered tentative.

To verify assignment of the chain direction of λP in our model, we performed a cross-linking mass spectrometric (CX-MS) analysis using the disuccinimidyl suberate (DSS) cross-linking agent (*Shi et al., 2014*; *Shi et al., 2015*). DSS has a maximum length of 11.4 Å, reacts with primary amines (eg. lysine side chains and the amino terminus) on proteins, and features a maximum reach threshold of ~30 Å (*Merkley et al., 2014*; *Shi et al., 2014*; *Shi et al., 2015*).

*E. coli* DnaB and the λP loader harbor 18 and 10 lysine residues, respectively, which allows for three types of crosslinks to be potentially observed: 1) intermolecular between DnaB and λP, 2) intra/intermolecular to DnaB, and 3) intra/intermolecular to λP. The CX-MS procedure provided a total 20 crosslinked peptides. 8 of these were in DnaB, eight between DnaB and λP, and four in λP. Notably, CX-MS is silent on whether the crosslinks in DnaB or λP arise from intramolecular or intermolecular interactions.

Evaluating our BP structural model against the CX-MS data is complicated by presence of six copies of DnaB and five copies of λP in the complex. Thus, a particular crosslinked peptide may arise from multiple instances of nominally equivalent, but distinct, lysine pairs; moreover, these pairs could arise from within or between subunits. Given these complications, we examined alpha carbon distances of all instances of pairs of lysine residues reported to be crosslinked by the CX-MS procedure; a crosslinked peptide was considered to be consistent with our model if the minimum distance between equivalent pairs of the underlying lysine residues was less than 30 Å. By this criterion, 16 of 20 CX-MS derived peptides were found to be consistent with the BP model. Notably, residues in λP associated with the six crosslinked peptides explained by the BP model mapped to the carboxy terminal ~60 residues of λP. Furthermore, the CX-MS data pointed to proximity of 3 positions on λP (K177, K200, and

K229) to the CTD of DnaB. One crosslink (DnaB 373 – λP 200) was not explained by our model as the distance (~43 Å) between the underlying pair of lysine residues exceeded the 30 Å reach threshold of the DSS agent. In addition, three other crosslinks (DnaB 2 – λP 229; DnaB 2 – DnaB 373; λP 2 and λ30) could not be evaluated as they involved residues that were not visible in our EM maps, and, thus, do not appear in our BP model. The CX-MS data support the chain polarity as featured in our model and precludes the possibility of the alternative polarity of λP (data not shown). A list of the CX-MS derived peptides and their relationship to our model appears in *Figure 3—figure supplement 1* and *Table 4*. The excellent agreement between the CX-MS reported crosslinks and our BP structural model confirms the finding of the binding study (*Figure 2—figure supplement 1*) and validates our choice of chain direction of the λP loader in the complex.

Collectively, orthogonal data (binding study, CX-MS) establish that the carboxy-terminal segment of λP is visualized in our EM map and establish the N to C direction of λP in the model. We note that, as part of the analysis, our BP model includes an implied assignment of the λP sequence to the structure; this assignment should be considered tentative.

## EM density corresponding to the amino terminal domain of λP

Inspection of our EM map at lower contours (four sigma in PyMol) reveals an additional region of EM density adjacent to the λP1 protomer (chain Z); this density likely represents this protomer's missing amino terminal domain. The quality of the density is comparatively lower than other parts of the map, nevertheless, we were able to position several alpha helical segments along with some intervening connections (*Figure 2—figure supplement 3E,F*). The finding that only a single λP protomer features extra density not only points to an unappreciated asymmetry in the BP complex, but also indicates presence of flexibility in the other λP NTD domains relative to their CTD domains in the BP complex.

We note that one of these helical segments appears to contact the CTD of subunit B, the subunit that lines the 'top' of the DnaB spiral, and lines the cracked interface of the helicase. This rod-like density could not be accounted by the known structure of DnaB, and distance measurements allow us to rule out that this density corresponds to the linker helix of chain A of DnaB. First, this segment of density does not pack against DnaB as expected for the linker helix; Second, for this segment to be the DnaB linker helix from chain A, it would have to be close enough to link to the cognate NTD. Our distance measurements to do not allow for this possibility. For example, linking the density in question to the appropriate segments would require linker spans of ~39 Å (before the linker helix) and ~66 Å (after the linker helix). However, these values exceed the maximum extent of the number of alpha carbons that could span this distance. For reference, the average distance for all DnaB protomers is ~23 Å and ~13 Å ahead of and behind the linker helix, respectively. In view of the above, we conclude that the linker helix and the accompanying linker segments of chain A are not visible on our maps. As such, we have modeled an alpha helix into the unaccounted density, and we tentatively conclude that this segment derives from the NTD of the P1 λP protomer in our model. In view of uncertainties associated with this segment, it does not appear in our PDB entry, but is shown in *Figure 2—figure supplement 3F*.

## Native mass spectrometry of the BP complex

The finding that five copies of the λP protein are bound to the DnaB-helicase was unanticipated and is at odds with prior measurements. Previous work implied a stoichiometry of $B_6P_3$ (*Mallory et al., 1990*), while more recent measurements suggest $B_6P_4$ (*Fok, 2002*). Moreover, efforts with the analogous DnaB•DnaC complex also resulted in disparate stoichiometry estimates, with values ranging from $B_6C_3$ (*Makowska-Grzyska and Kaguni, 2010*) to $B_6C_6$ (*Kobori and Kornberg, 1982*; *Galletto et al., 2003*). A 25 Å map of the DnaB•DnaC complex obtained from a negative stain EM analysis shows density for six DnaC monomers (*Arias-Palomo et al., 2013*). However, Kaguni et al. imply that the stoichiometry is

$B_6C_3$ when DnaB is a part of the DnaA-DnaB-DnaC pre-priming origin complex (*Makowska-Grzyska and Kaguni, 2010*).

To better understand the stoichiometry of the BP complex, we used native mass spectrometry (MS) to obtain accurate mass estimates. Our measurements indicate that the predominant species in our BP sample preparation exhibits a mass of 446.5 kDa ($B_6P_5$); additional species with masses of 473.1 kDa ($B_6P_6$) and 419.9 kDa kDa ($B_6P_4$) were also observed, though at lower intensities (*Figure 3C* and *Table 3*). In comparison to the stoichiometric heterogeneity seen with our preparations of the BP complex, addition of a 13.1 kDa 43-mer ssDNA derived from the Oriλ phage replication origin revealed a single entity with a mass of 459.5 kDa; this mass corresponds to the $B_6P_5$ helicase loader complex bound to ssDNA (*Figure 3D* and *Table 3*). Moreover, elimination of stoichiometric heterogeneity in BP complexes is specific to the DNA sequence employed. For example, inclusion of thymidine homopolymers of varying nucleotide length (eg. $T_{25}$, $T_{35}$, and $T_{45}$) showed no evidence of a protein-DNA complex, nor was the heterogeneity in subunit stoichiometry eliminated (*Figure 3—figure supplement 2*). Two aspects of this finding are remarkable. First, isolated DnaB-helicase is not known to show preference for particular DNA sequences, and second, isolated DnaB-helicase is known to bind thymidine homopolymers, amongst other ssDNA sequences (*Bujalowski and Jezewska, 1995*; *Jezewska and Bujalowski, 1996*; *Jezewska et al., 1996*). Thus, the nature of the contacts to ssDNA have radically changed: from non-specific in the isolated helicase to exhibiting a specificity in the loader complex for sequences derived from the replication origin. These preliminary data hint that DnaB, while complexed to the loader, may make few, if any, contacts to ssDNA. Second, our findings point to the possibility that contacts to origin-derived ssDNA might be predominantly, or even exclusively, made by the λP loader.

Both the BP and the BP-ssDNA complex samples were prepared in the presence of an excess of nucleotide. However, the procedure for preparing samples for native MS includes a step that removes nucleotides; this is done to reduce the extent of non-specific adduction of nucleotide to the entities of interest. In this context, we note that the measured masses for both the BP and the BP-ssDNA complex indicates that no nucleotide is bound to either complex. We infer from this observation that nucleotide is not necessary for stability of the BP complex. A parallel native MS analysis of the *Aquifex aeolicus* DnaB•DnaC complex reveals the presence of five ATP molecules, though not whether they are bound by DnaB or DnaC or some combination thereof (below); the differing results are likely due to a varying affinity for nucleotides by the associated complexes.

The observation that ssDNA derived from the oriλ replication origin eliminates subunit heterogeneity and stabilizes the $B_6P_5$ entity leads us to conclude that the $B_6P_6$ and $B_6P_4$ species present in our preparations are likely unstable intermediates or artifacts of the recombinant method used to prepare the complex. Taken together, our structural and biophysical analyses firmly establish that the stoichiometry of the opened BP complex required for initiation of replication is DnaB: 6 and λP: 5. Notably, our stoichiometry estimate is entirely consonant with disposition of the CTDs in the BP complex. Of the six possible CTD interfaces that could provide binding sites for λP, only five are intact; the sixth interface has been ruptured owing to the open spiral configuration of the CTDs in the complex, and, thus, cannot productively bind a sixth λP.

## Analysis of the structure of the λ P helicase loader

Comparative structural analysis of the carboxy-terminal domain of the λP helicase loader against the PDB structure database using the Dali (*Holm and Rosenström, 2010*), PDBefold (*Krissinel and Henrick, 2004*), and Phyre (*Kelley et al., 2015*) tools revealed no close structural relatives. Notably, λP bears no structural similarity with the DnaC helicase loader (*Mott et al., 2008*), a finding anticipated by their divergent amino acid sequences (*Nakayama et al., 1987*).

## Calculation of helical parameters for the CTD and NTD tiers in the BP complex

In the BP complex, the NTD and CTD tiers were found in an open right handed pseudo-helical configuration. To analyze these configurations, we employed the language of helical parameters commonly used to describe protein and DNA helices (*Lu and Olson, 2003*). Specifically, our analysis encompasses three parameters: (1) helical pitch, which refers to the length of one complete helical turn along the helix axis, (2) helical twist, which refers to the angle made by successive helical subunits parallel to the helix axis, and (3) helical inclination, which refers to rotation of the helical subunit along an axis perpendicular to the helical axis. The above parameters provide convenient handles for comparing the DnaB pseudo-helix in the BP complex to other pseudo-helical DnaB configurations.

To enable calculation of the above helical parameters, both the principal components of the closed planar forms were aligned to the Cartesian X, Y, and Z axes using the 'orient' command in PyMol (*Schrodinger LLC, 2017*). The efficacy of this operation was verified by inspecting the Z coordinate of the center of mass of each domain (CTD, NTD). If aligned, the Z coordinates should lie on a plane perpendicular to the Z- axis; this is indeed the case. Subsequent superpositions aligns the pseudo-helical axes of the comparison DnaB tiers to the Z axis.

The right-handed pseudo-helical parameters of the CTD sub-structures of the DnaB helicase were determined by first aligning the appropriate portion of the three principal inertial axes of the closed planar constricted hexameric form of DnaB (PDB = 4NMN) to the Cartesian X, Y, and Z axes (PyMol or MOLEMAN2 (*Kleywegt and Jones, 1997*; *Kleywegt, 2007*)). In a second step, the DnaB components of the ssDNA complex (PDB = 4ESV) and the BP complex were aligned to one subunit of the planar closed constricted DnaB hexamer (PDB = 4NMN). The rise per DnaB subunit was calculated by subtracting the Z coordinate from the center of masses of the spiral and planar forms (eg. 4ESVChainA•CTD – 4NMNChainA•CTD, BPChainA•CTD – 4NMNChainA•CTD, etc); this procedure was repeated for each subunit. The helical twist value of each subunit around the pseudo-helical axis was calculated using LSQMAN (*Kleywegt and Jones, 1997*; *Kleywegt, 2007*). The average helical twist value excluded DnaB subunit B, which is located adjacent to the ruptured interface at the top of the helical spiral; this subunit displays a twist of ~78° and represents an outlier. Finally, the helical pitch was obtained from the quotient of the rise per subunit and the rotation per subunit.

The helical parameters for the spirals formed by the NTD tiers of various forms DnaB were calculated as above, except for two changes. First, the planar dilated form of DnaB (PDB = 2R6A) was used as a reference for the ssDNA bound complex (PDB = 4ESV) and the planar constricted form (PDB = 4NMN) was used as a reference for the BP complex. Second, parameters were calculated using the only the globular domain (residues 31–113) of each NTD.

Helical inclination values were calculated by taking all possible pairs of subunits of a DnaB entity, superimposing the first member of the pair on the first pair of a reference pair of subunits from the dilated closed planar form, and then calculating the rotation required to bring the second member of the comparison pair into alignment with the second subunit of the reference pair.

## Sequence and structural conservation of the DnaB helicase

The DnaB helicase is an essential and highly conserved bacterial protein; conservation is reflected in the close structural correspondence of various orthologs (*Figure 5—figure supplement 1*). *E. coli* DnaB helicase is a member of the TIGR00665 conserved protein domain family; analysis of 154 sequence representatives of this family reveals an overall sequence similarity of 63%, where equivalence between amino acids at a given position was established using a normalized BLOSUM62 substitution matrix. Close inspection of the alignment indicates that amino-terminal (NTD: 1–151) and linker (152–202) domains are less

conserved in sequence than the carboxy-terminal (CTD: 203–471) domain. A prominent patch of high sequence conservation lines the DNA binding channel of the opened helicase. Similar results were obtained using the ConSurf server (*Ashkenazy et al., 2016*).

In every structure of the DnaB helicase, the CTDs exhibit a pseudo-six-fold arrangement, either in a planar or spiral configuration. By contrast, the six N-terminal domains of the NTD layer feature a trimer of dimers configuration that displays pseudo-three-fold symmetry. This arrangement is enabled by adoption of an alternate conformation by every other NTD domain in the layer. Termed cis and trans ((*Wang et al., 2008*), also inner NTD and outer NTD (*Bailey et al., 2007a*)), the alternately configured monomers differ on the relative orientation of the NTD and CTD. The NTD of a *cis* monomer features a helical hairpin substructure that points toward the CTD of the same chain; by contrast, in the *trans* monomer, this sub-structure points away from the parent CTD (*Wang et al., 2008*). Alternate conformations of DnaB monomers brings two NTDs into a dimeric relationship, and three such dimers associate to form the complete NTD layer, and provide its distinctive triangular shape.

Although four divergent configurations for the hexameric DnaB helicase have been described (closed-planar-dilated, closed-planar-constricted, closed-spiral, and open spiral (the instant structure), the structures of component domains of DnaB are highly conserved. 12 instances of the NTD monomer sub-structure of DnaB (PDB entries: 1B79, 2Q6T, 2R5U, 2R6A, 2VYF, 3BGW, 3GXV, 4ESV, 4M4W, 4NMN, 4ZC0, and the DnaB• λP complex) have been described. The NTD encompasses ~150–170 residues of which residues ~ 1–100 form a globular domain and residues ~ 100–170 fold into an anti-parallel helical bundle, giving the NTD the overall appearance of a golf club. The extent of structural conservation of the NTD monomer is revealed by an RMSD value of 2.0 ± 0.85 Å exhibited by the alignment of the 12 available sub-structures.

The four divergent DnaB hexameric configurations exhibit three distinct NTD dimer interfaces (*Wang et al., 2008*; *Strycharska et al., 2013*), termed tail-to-tail (between the helical hairpin components of two NTDs, one belonging to a *cis* DnaB monomer and the other belonging to a *trans* monomer), head-to-head (between globular domains of two NTDs), and head to tail (seen only in the constricted form) (*Figure 7*). Of these, the tail to tail interface buries the most surface area, and is presumed to be the most stable. Superposition analysis suggests conservation of each of these interfaces, though some plasticity exists among the various DnaB entities (*Figure 5—figure supplement 1*).

The CTD of DnaB spans ~280 residues and resembles a sphere. The CTD is also the locus of highest sequence conservation, and this is reflected in the close structural correspondence of the various instances in the database. Superposition analysis of the 12 instances of the CTD (PDB entries: 1CR0, 1EOK, 2Q6T, 2R6A, 3BH0, 3BGW, 4A1F, 4ESV, 4M4W, 4NMN, 4ZC0, and the DnaB• λP complex) reveals an RMSD value of 1.31 Å ±0.34 Å. However, the four known configurations of the DnaB hexamer feature distinct arrangements of individual CTDs as described above and in (*Bailey et al., 2007a*; *Wang et al., 2008*; *Itsathitphaisarn et al., 2012*; *Strycharska et al., 2013*).

In summary, the DnaB helicase is a highly dynamic entity whose arrangement of component domains varies with the configuration of the hexamer, but whose domain sub-structures exhibit high structural conservation.

## Comparison to other bacterial helicase • helicase loader complexes

Analysis of other helicase • helicase loader entities, in combination with the present work, suggests that bacteria have evolved diverse mechanisms to mediate assembly of the replicative helicase (*Figure 8—figure supplement 1*). Elements of these bacterial helicase assembly paradigms can also be found in the more elaborate replication initiation systems in eukaryotes (*Bleichert et al., 2017*). Helicase loading factors present in at least two flavors (*Davey and O'Donnell, 2003*; *Soultanas, 2012*). In the first instance are the ring-breakers, as exemplified by *E. coli* DnaC, which crack open the DnaB helicase to enable assembly around DNA (*Davey and O'Donnell, 2003*; *Arias-Palomo et al., 2013*). Our work implies that the phage λP loader is also a ring-breaker. Second, are the ring-makers, for example, *G. subtilis*

DnaB/DnaI, which appear to assemble the replicative helicase from monomers (*Velten et al., 2003*; *Soultanas, 2012*; *Bell and Kaguni, 2013*). Structural analysis of the *G. stearothermophilus* hexameric helicase bound to the *G. subtilis* DnaI loading factor and the DnaG primase revealed a closed helicase ring; this finding provides support for a potential ring-making role for DnaI (*Liu et al., 2013*).

This work, and that of many other groups (*Kobori and Kornberg, 1982*; *Mallory et al., 1990*; *Learn et al., 1997*; *Stephens and McMacken, 1997*; *Davey et al., 2002*; *Davey and O'Donnell, 2003*; *Strycharska et al., 2013*; *Chodavarapu et al., 2015*; *Felczak et al., 2017*), has established that, although both DnaC and λP are ring-breaking helicase loaders, they operate via mechanisms that are similar, but also different. Both are essential proteins (*Wechsler, 1975*; *Echols and Murialdo, 1978*). Both complex with *E. coli* DnaB helicase, and indeed, their binding sites overlap (this work, (*Mallory et al., 1990*; *Chodavarapu et al., 2015*)). Moreover, both proteins inhibit the ATPase and, concomitantly, the helicase activity of DnaB; both deliver the helicase to cognate origins; both bind to ssDNA and influence binding of DNA by DnaB (*Wahle et al., 1989a*; *Wahle et al., 1989b*; *Mallory et al., 1990*; *Learn et al., 1997*; *Davey et al., 2002*); both require eviction of the loader for activation of the helicase. However, DnaC and λP are completely unrelated in amino acid sequence and structure, and their eviction from the origin complex proceeds via distinct mechanisms. For DnaC, ATP dynamics in its AAA +ATPase domain (*Wahle et al., 1989a*; *Davey et al., 2002*; *Mott et al., 2008*; *Bell and Kaguni, 2013*), along with other protein binding events (*Makowska-Grzyska and Kaguni, 2010*; *Chodavarapu et al., 2015*) are significant features in its eviction from the origin complex. By contrast, λP neither binds nor hydrolyzes ATP (*Biswas and Biswas, 1987*); rather, eviction, and concomitant helicase activation, require engagement of the host DnaK/DnaJ/GrpE chaperone machinery to partially unfold the λP protein (*LeBowitz et al., 1985*; *Mensa-Wilmot et al., 1989*; *Wyman et al., 1993*; *Polissi et al., 1995*).

The diversity in biochemical mechanisms is also reflected in the structures of the various complexes (*Figure 8—figure supplement 1B*). The low-resolution structure of the *E. coli* DnaB•DnaC (BC) complex (*Arias-Palomo et al., 2013*) shows an arrangement of NTD and CTD layers that resemble those seen in the BP complex (a high resolution comparison must await availability of an atomic model of the BC complex). Although there is overlap in the binding sites on the helicase for the two loaders, the stoichiometry and disposition of the helicase loader proteins differ substantially between the two complexes. First, the BP complex harbors five copies of the loader whereas six copies of DnaC reside in the BC complex. To provide further context for the unanticipated stoichiometry differences between the two loader complexes, we extended our native MS experiments to the *Aquifex aeolicus* DnaB•DnaC (BC) ensemble. These experiments revealed a single entity of mass = 485.3 kDa; this mass corresponds to a stoichiometry of $B_6C_6$ (*Figure 8—figure supplement 1*) and includes 5 ATP molecules. The measured stoichiometry is in agreement with prior structural and biochemical analyses of the *E. coli* entity (*Kobori and Kornberg, 1982*; *Galletto et al., 2003*; *Arias-Palomo et al., 2013*). Second, each λP loader binds at a DnaB subunit interface, and, consequently, contacts two subunits. By contrast, in the BC complex, each DnaC molecule appears to contact a single DnaB chain in the hexamer, although hydrogen deuterium exchange mass spectrometry (HDX-MS) experiments point to additional contacts (*Chodavarapu et al., 2015*), including the possibility of contacts to an adjacent subunit. Furthermore, each loader employs a distinct segment to complete an extended binding site on DnaB. In addition to the compact helical domain, λP sends out its carboxy terminal segment to complete its extended binding site. By contrast, DnaC, in addition to contacts mediated by the AAA+ domain, deploys its amino terminal 75 residues to complete the binding site with DnaB (*Arias-Palomo et al., 2013*; *Chodavarapu et al., 2015*). Indeed, the HDX-MS analysis of this interaction predicts that DnaC will contact two regions of DnaB (295–304 and 431–435) (*Chodavarapu et al., 2015*). Notably, both predicted areas lie within the interface between DnaB and λP. Not only does this result provide support for a shared binding site between the two ring breakers DnaC and λP, but also highlights how bacteria and

bacteriophage λ converged on similar helicase opening mechanisms, but which are implemented distinctly by proteins entirely unrelated in sequence or structure.

