## [Decision Letter]

Thank you for submitting your article "Mechanisms of Opening and Closing of the Bacterial Replicative Helicase" for consideration by *eLife*. Your article has been reviewed by three peer reviewers, including Michael R Botchan as the Reviewing Editor and Reviewer #1, and the evaluation has been overseen by Philip Cole as the Senior Editor. The following individual involved in review of your submission has agreed to reveal their identity: Huilin Li (Reviewer #3). A further reviewer remains anonymous.

The reviewers have discussed the reviews with one another and the Reviewing Editor has drafted this decision to help you prepare a revised submission.

The reviewers all found this manuscript interesting and the conclusions well justified. The contribution is captured by the following summary.

The loading of replicative helicases onto DNA, often assisted by dedicated loader proteins, is an essential step during the initiation of DNA replication. In the manuscript "Mechanisms of Opening and Closing of the Bacterial Replicative Helicase", Chase et al., determine a cryo-EM structure of the *E. coli* replicative helicase, DnaB, in complex with the helicase loader protein from bacteriophage λ at an average resolution of ~4.1 A. Surprisingly, the structure shows that the DnaB hexamer binds 5 copies of the phage loader protein at its C-terminal side, with each λ P associating with two adjacent DnaB protomers. This unusual stoichiometry is supported by native mass spectrometry. In the BP complex structure, the DnaB hexamer adopts a constricted and spiral conformation that leads to a break in both the NTD and CTD tiers of the DnaB ring. This gap is wide enough to allow ssDNA to enter the central cavity and therefore supports a ring-breaking mechanism for loading of the *E. coli* replicative helicase. The authors further note that the DnaB ATPase ring is held in a restrained configuration that prevents the DNA binding loops and ATPase centers to adopt a conformation competent for ssDNA binding and ATP hydrolysis, respectively. However, this conformation likely is in some equilibrium with other states (see below for an issue that should be addressed in revisions). Overall these observations lead the authors to propose detailed models for both ring opening and closing of DnaB during replication initiation.

Several crystal structures of DnaB have been determined previously, either in isolation or in complex with ssDNA, but structural information of DnaB-loading intermediates has been limited to a 25 A structure of DnaB in complex with its cognate *E. coli* loader protein, DnaC. The structure of the BP complex determined by Chase et al. thus represents the first near-atomic resolution structure of a DnaB-loader complex in the pre-loading state. Although the conformation of DnaB in the BP complex structure is very reminiscent of its configuration in the DnaBC complex, the new structure allows the authors to refine existing models for DnaB ring opening and closing and illustrates how two distinct loader proteins can remodel DnaB in a similar manner.

Suggested revisions:

1) The authors find that the inclusion of ssDNA improves the homogeneity of the complex when analyzed by native mass spectrometry and that it eliminates low abundant B6-P4 and B6-P6 assemblies. Moreover, it seems logical to assume that the use of a non-hydrolyzable nucleotide instead of ATP (which is hydrolyzed to ADP in the map) might stabilize the complex further. It is unclear why the authors did not attempt either approach, which may provide a higher resolution structure and would allow for more confident modeling of the λ P protein. Further explanation of the authors' reasoning would be appreciated.

2) The conformation of DnaB in the BP complex seems very similar to its conformation in the *E. coli* DnaB-DnaC complex. Although the authors acknowledge this in the main text, it would be useful to include this comparison in figure form as well, both for the ATPase domains and the NTDs.

3) In Figure 2—figure supplement 1D, the authors show co-purification of untagged DnaB with the His-tagged C-terminal domain of λ P (aa 103-233) to support that the C-terminal region of λ P binds DnaB. However, this biochemical experiment lacks important controls, including a) a pull-down performed in the absence of His-tagged λ P protein to control for unspecific association of DnaB with beads and b) a pull-down using a tagged N-terminal fragment of λ P, which should not associate with DnaB according to the authors' interpretation. Moreover, the full-length λ P protein and the C-terminal fragment 103-233 both migrate around 20kDa in the gel despite their significantly different molecular weights. It would be helpful if the authors commented on this discrepancy.

4) It is very difficult to identify individual particles in Figure 2—figure supplement 2D. The authors should consider showing a micrograph with better signal to noise and only highlight a few picked particles by circles. In addition, the 8 volumes that resulted from the 3D classification should be shown. There is also a discrepancy in the reported resolution of the cryoET reconstruction (Figure 2—figure supplement 2B) in the legend and text (40 vs 25 A), which should be clarified.

5) To demonstrate agreement of the built model with the EM map, the authors should provide the FSC comparing model with EM map.

6) Figure 2—figure supplement 5: It is unclear what the authors mean by "heterogeneous" and "homogenous" refinement and, therefore, these approaches should be explained.

7) Figure 6B: The authors' conclusion that the ATPase sites in the BP complex are misaligned is only weakly supported by the data shown. The authors report movements with distances much lower than the resolution of the structure. Although the authors acknowledge that the reported distances are only tentative, this practice is problematic, and we recommend that the authors should remove these distance measurements. In addition, it would be important to show densities for the side chains in the BP complex at the ATPase sites to demonstrate that the positioning of these side chains in the BP model is reliably supported by density in the map. Likewise, the authors should show densities for the bound nucleotide to support their conclusion that five ATPase sites are occupied by ADP instead of ATP.

8) A reviewer was confused about the exact state of the Stage II complex. Because Stage III is the loader-origin-helicase complex, so, Stage II as described in this manuscript should be competent and ready to bind the origin DNA. But in several places, it was stated that the structure does not hydrolyze ATP (ATP binding sites are disrupted) and cannot bind ssDNA (Figure 6) (subsection “The DNA and ATP binding sites are disrupted in the DnaB• λP complex”). This seems to conflict with the mass spec data indicating the structure's ability to bind a 43-mer ori-lamda ssDNA (Figure 3D), and the fact that five ADP molecules occupy the structure – meaning the structure has hydrolyzed ATP. This issue is alluded to in the summary- with a potential resolution? If the reported structure is indeed incompatible with DNA binding, does this mean the structure is a non-functional structure not found in a cell or perhaps a minor conformation in the cell?

9) Related to the point in 8), the native mass spec shows the formation of DnaB-loader P-43-mer DNA (Figure 3D). Did the authors attempt to visualize the structure, and if yes, is DNA visible, and is the structure distinct from the one reported here which is in the absence of DNA?

10) Native mass spec revealed three species 6:5 (dominant), 6:4, and 6:6 stoichiometric binding. Did the authors identify the two non-dominant species (B6P4 and B6P6) in their 3D classification scheme of the cryo-EM dataset?

---

## [Author Response]

Suggested revisions:1) The authors find that the inclusion of ssDNA improves the homogeneity of the complex when analyzed by native mass spectrometry and that it eliminates low abundant B6-P4 and B6-P6 assemblies. Moreover, it seems logical to assume that the use of a non-hydrolyzable nucleotide instead of ATP (which is hydrolyzed to ADP in the map) might stabilize the complex further. It is unclear why the authors did not attempt either approach, which may provide a higher resolution structure and would allow for more confident modeling of the λ P protein. Further explanation of the authors' reasoning would be appreciated.

Both the BP and the BP-ssDNA complex samples were incubated in the presence of large excess of nucleotide (ATP). Immediately prior to taking the MS spectra, the samples were buffer exchanged into a volatile salt that does not include nucleotide. Nucleotide is excluded from this buffer because we found that this reduced the extent of non-specific adduction to the entities of interest and provided cleaner spectra. Inclusion of nucleotide during buffer exchange led to nonspecific binding and broadening of the MS peaks, which complicated their interpretation. From our native MS experiments, we infer that although nucleotide may play a role in formation of the BP complex, it does not appear necessary for stability of the BP complex. The finding that the BP complex is stable without nucleotide (both in the absence and presence of ssDNA) led us not to prioritize cryoEM experiments with non-hydrolysable ATP analogues. However, this will be something we pursue in the future.

In parallel, we performed native MS measurements with the distinct *Aquifex aeolicus* DnaB-DnaC helicase – helicase loader (BC) complex (Figure 8—figure supplement 1 panel A). The resulting mass estimates provide an important context for our efforts with the BP complex because the BC complex was treated in exactly the same way. In contrast to the BP complex, mass estimates for the BC complex reveal presence of five ATP molecules. From the comparison of the MS results of two samples, we conclude that the buffer exchange procedure differentially removes nucleotide molecules based on the individual complex’s affinities.

2) The conformation of DnaB in the BP complex seems very similar to its conformation in the E. coli DnaB-DnaC complex. Although the authors acknowledge this in the main text, it would be useful to include this comparison in figure form as well, both for the ATPase domains and the NTDs.

The published structural analysis of the DnaB-DnaC complex is a 25 Å negative stain EM analysis from the group of James Berger (Arias-Palomo, 2013). Although fits of previously determined structures of DnaB and DnaC to the EM map are discussed in the published work, no atomic model of the DnaB-DnaC complex has been deposited in the structural database for EMDB entries 2321 and 2322. As such, there are no published coordinates for the DnaB-DnaC complex against which to compare our atomic model for the DnaB-LP complex.

We have adjusted the narrative to reflect the above.

3) In Figure 2—figure supplement 1D, the authors show co-purification of untagged DnaB with the His-tagged C-terminal domain of λ P (aa 103-233) to support that the C-terminal region of λ P binds DnaB. However, this biochemical experiment lacks important controls, including a) a pull-down performed in the absence of His-tagged λ P protein to control for unspecific association of DnaB with beads and b) a pull-down using a tagged N-terminal fragment of λ P, which should not associate with DnaB according to the authors' interpretation. Moreover, the full-length λ P protein and the C-terminal fragment 103-233 both migrate around 20kDa in the gel despite their significantly different molecular weights. It would be helpful if the authors commented on this discrepancy.

To address the important missing control of whether the amino-terminal segment of λ P also binds to DnaB, we tested a construct that encompassed λ P residues 1 to 110 (with either a CTerminal or N-Terminal histidine tag) for co-expression and binding studies with DnaB. Multiple efforts to express this segment of λ P, in isolation or when co-expressed with DnaB, failed to show any expression of truncated λ P whatsoever. However, during co-expression, DnaB was clearly expressed; this observation suggests that our expression trials succeeded, but that the λ P construct of interest failed to express. As such, we conclude that this segment of λ P is unstable under the conditions tested. We have adjusted the text to reflect that this important control experiment could not be performed for technical reasons.

We have corrected the labeling of the gels in Figure 2—figure supplement 1 to reflect the molecular masses for full-length and the C-terminal construct of λ P. This entailed replacing the original gel for panel C of Figure 2—figure supplement 1 with one that better depicted the molecular masses of DnaB and λ P. The corrected figure now shows that full length λ P migrates at ~27 kDa, whereas the histidine tagged C-terminus of λ P (residues 103-233) migrates around ~15 kDa.

**Author response image 1. respfig1:** Untagged DnaB • λP does not bind nonspecifically to NiNTA beads.

To address the question of non-specific binding of untagged DnaB•λP to NiNTA beads, we performed the experiment analyzed in Author response image 1. Untagged DnaB•λP complex was mixed with beads and subjected to washes with progressively higher concentrations of imidazole. Figure 1 shows that the untagged BP complex is washed off the column at concentrations of imidazole corresponding to weak binding. By contrast, histidine tagged λP (103-233) elutes from the NiNTA beads at a concentration of 250 mM; this interaction is considered strong (panel D of Figure 2—figure supplement 1). The experiment in Figure 1 above and in panel D of Figure 2—figure supplement 1 were performed under the same conditions. From Figure 1, we conclude that the untagged BP complex shows little/no specific interaction with the NiNTA beads. As such, the interaction between tagged λP (103-233) and untagged DnaB documented in panel D of Figure 2—figure supplement 1 is specific and robust. We have adjusted the text to reflect that outcome of this control experiment.

4) It is very difficult to identify individual particles in Figure 2—figure supplement 2D. The authors should consider showing a micrograph with better signal to noise and only highlight a few picked particles by circles. In addition, the 8 volumes that resulted from the 3D classification should be shown. There is also a discrepancy in the reported resolution of the cryoET reconstruction (Figure 2—figure supplement 2B) in the legend and text (40 vs 25 A), which should be clarified.

We agree that the micrograph shown in Figure 2—figure supplement 2 is difficult to identify individual particles. To remedy this issue, we have: (1) changed the displayed micrograph to one with higher contrast, and (2) elected to circle particles identified by template-based particle picking on the top half of the micrograph only, while leaving the bottom half of the micrograph unobstructed.

To address the question of the 8 volumes from the 3D classification, we have created an additional figure (Figure 2—figure supplement 4). This figure depicts how we used the 3D classification procedure to clean our particle stack, as described in the Material and methods section and Appendix.

On the question of the resolution of the cryoET volume; this is challenging to determine due to the relatively low number of particles used to generate the model (1,000 particles were used to generate 3 cryoET volumes, each of which was nearly identical regarding global architecture). We estimate by visual comparison to filtered maps that the cryoET volume is in the 30-40 Å range and have edited the text in the manuscript to reflect this. Our use of cryoET to generate a low resolution model of the BP complex is described here (Noble et al., 2018). All instances of the reported resolution have been set to the same value.

5) To demonstrate agreement of the built model with the EM map, the authors should provide the FSC comparing model with EM map.

We have addressed this by using the Mtriage functionality within the Phenix software suite to compare our cryoEM map to our model. We have reported the data in a FSC plot in panel B of Figure 2—figure supplement 3. The resulting FSC comparing the model with our EM map results in two independently refined half-sets converging at 4.1 Å; this value is in agreement with the resolution of our map.

6) Figure 2—figure supplement 5: It is unclear what the authors mean by "heterogeneous" and "homogenous" refinement and, therefore, these approaches should be explained.

We have explained and defined these approaches more clearly in the Appendix of this manuscript (section titled “Single-Particle CryoEM and CryoET Analysis of the BP Complex). We have also altered panels A and B of Figure 2—figure supplement 5 to more clearly represent this procedure. Heterogenous refinement strategies involve the use of multiple ab initio models which each class is refined against, whereas homogenous refinement uses a single ab initio model for initialization of six classes.

7) Figure 6B: The authors' conclusion that the ATPase sites in the BP complex are misaligned is only weakly supported by the data shown. The authors report movements with distances much lower than the resolution of the structure. Although the authors acknowledge that the reported distances are only tentative, this practice is problematic, and we recommend that the authors should remove these distance measurements. In addition, it would be important to show densities for the side chains in the BP complex at the ATPase sites to demonstrate that the positioning of these side chains in the BP model is reliably supported by density in the map. Likewise, the authors should show densities for the bound nucleotide to support their conclusion that five ATPase sites are occupied by ADP instead of ATP.

We have removed the distance measurements but left the red arrows to reflect our best estimate for how the hairpin sub-structure has become mis-aligned at the level of the associated α carbon of residues known to be important for nucleotide hydrolysis (K440 and R442).

We have added Figure 2—figure supplement 5 (panel B) that displays densities for the five nucleotide binding sites populated with ADP. These illustrations include insets of density that corresponds to the five observed ADP molecules.

8) A reviewer was confused about the exact state of the Stage II complex. Because Stage III is the loader-origin-helicase complex, so, Stage II as described in this manuscript should be competent and ready to bind the origin DNA. But in several places, it was stated that the structure does not hydrolyze ATP (ATP binding sites are disrupted) and cannot bind ssDNA (Figure 6) (subsection “The DNA and ATP binding sites are disrupted in the DnaB• λP complex”). This seems to conflict with the mass spec data indicating the structure's ability to bind a 43-mer ori-lamda ssDNA (Figure 3D), and the fact that five ADP molecules occupy the structure – meaning the structure has hydrolyzed ATP. This issue is alluded to in the summary- with a potential resolution? If the reported structure is indeed incompatible with DNA binding, does this mean the structure is a non-functional structure not found in a cell or perhaps a minor conformation in the cell?

In the introduction, we have clarified that in Stages II and III the ATPase and ssDNA translocation activities are suppressed in the helicase – helicase loader complex, but that the complex retains the ability to bind ssDNA. As is noted above, our native MS data reveals that the BP complex binds to λ phage origin derived ssDNA, but not to other non-origin sequences of comparable length (Figures 3—figure supplement 2). However, the native MS data is silent on which entity in the complex is actually making contacts to ssDNA. DnaB is not known to show a preference for any ssDNA sequence, and, as such, it is striking that on binding the loader, the complex now exhibits a preference for origin derived ssDNA. From the appearance of the ssDNA binding site on DnaB, we infer that binding by DnaB may be severely disrupted, if not completely abolished. In the discussion, we have clarified that, taken together, our data imply that most, if not all, of the contacts to origin derived ssDNA may be made by the λP loader and that this finding sheds light on the nature of Stage III. As well, we propose that the structure captured by our EM analysis in indeed a physiological conformer, and that it might perform a previously uncharacterized activity: an origin specificity factor for the DnaB helicase.

9) Related to the point in 8), the native mass spec shows the formation of DnaB-loader P-43-mer DNA (Figure 3D). Did the authors attempt to visualize the structure, and if yes, is DNA visible, and is the structure distinct from the one reported here which is in the absence of DNA?

Structural analysis of the BP-origin ssDNA complex is ongoing in the Jeruzalmi research group. At this time, our efforts are too preliminary for discussion.

10) Native mass spec revealed three species 6:5 (dominant), 6:4, and 6:6 stoichiometric binding. Did the authors identify the two non-dominant species (B6P4 and B6P6) in their 3D classification scheme of the cryo-EM dataset?

We have exhaustively probed our cryoEM dataset for additional stoichiometries of the BP complex, and have not found evidence of the B_6_P_4_ or B_6_P_6_ complexes suggested to be present in our preparation by our native MS experiments. In addition to the B_6_P_5_ volumes that we have discussed in the manuscript, we find evidence for a) incomplete DnaB rings or b) non-planar, cracked DnaB absent density corresponding to λ P. Our working explanation for this is that either these two populations (B_6_P_4_ or B_6_P_6_) are not stable enough to survive the plunge freezing process during grid preparation or that these populations were artifacts associated with the method of complex preparation. These efforts are detailed in the Appendix.